# Learning from Demonstrations via Capability-Aware Goal Sampling

**Yuanlin Duan**
Rutgers University
yuanlin.duan@rutgers.edu

**Yuning Wang**
Rutgers University
yw895@cs.rutgers.edu

**Wenjie Qiu**
Rutgers University
wq37@cs.rutgers.edu

**He Zhu**
Rutgers University
hz375@cs.rutgers.edu

## Abstract

Despite its promise, imitation learning often fails in long-horizon environments where perfect replication of demonstrations is unrealistic and small errors can accumulate catastrophically. We introduce Cago (Capability-Aware Goal Sampling), a novel learning-from-demonstrations method that mitigates the brittle dependence on expert trajectories for direct imitation. Unlike prior methods that rely on demonstrations only for policy initialization or reward shaping, Cago dynamically tracks the agent's competence along expert trajectories and uses this signal to select intermediate steps—goals that are just beyond the agent's current reach—to guide learning. This results in an adaptive curriculum that enables steady progress toward solving the full task. Empirical results demonstrate that Cago significantly improves sample efficiency and final performance across a range of sparse-reward, goal-conditioned tasks, consistently outperforming existing learning from-demonstrations baselines.

## 1 Introduction

Imitation Learning (IL) provides a powerful paradigm for training agents using expert demonstrations, effectively alleviating the exploration challenges common in Deep Reinforcement Learning (DRL) (Arulkumaran and Lillrank, 2024). The simplest form of IL is Behavior Cloning (BC), which directly supervises policy actions based on the states visited by the expert (Bain and Sammut, 1995; Torabi et al., 2018). However, BC often suffers from compounding errors when the learned policy deviates from expert trajectories. To overcome this limitation, modern approaches such as GAIL (Ho and Ermon, 2016), PWIL (Dadashi et al., 2020), and AdRIL (Eysenbach et al., 2021) seek to align the state–action distributions of the agent and the expert through adversarial or distribution-matching objectives. In parallel, Inverse Reinforcement Learning (IRL) methods (Ziebart et al., 2008) aim to infer underlying reward functions from demonstrations, which can then guide reinforcement learning. More recently, advances in offline and offline-to-online RL, such as CQL (Kumar et al., 2020) and Cal-QL (Nakamoto et al., 2023), integrate demonstrations as anchors to regularize policy learning. These methods penalize value estimates that diverge from demonstrated behavior, mitigating overestimation and instability caused by out-of-distribution actions.

However, existing IL methods often struggle with complex, long-horizon tasks because they fail to reason about which parts of the task the agent has already mastered and which remain challenging. In particular, distribution-matching approaches perform flat matching—attempting to align occupancy measures over the entire trajectory distribution without considering the agent's evolving capabilities. This leads to poor exploration guidance, especially in the early stages of training when the agent

39th Conference on Neural Information Processing Systems (NeurIPS 2025).

seldom reaches meaningful parts of the state space. As a result, the learned reward function tends to assign uniformly low rewards, yielding uninformative gradients and hindering effective policy improvement. Some prior work proposes demonstration-guided curriculum learning that trains agents to solve tasks by starting near the goal or high-reward states and gradually expanding to earlier parts of the trajectories (Resnick et al., 2018; Salimans and Chen, 2018; Tao et al., 2024). However, these approaches rely on the ability to reset the agent to arbitrary demonstration states—an assumption impractical in real-world settings due to challenges in replicating physical conditions like joint velocities and angular momentum.

We propose Cago (Capability-Aware Goal Sampling), a new learning-from-demonstrations framework that explicitly aligns the agent's learning process with its evolving capabilities. Unlike prior methods that use demonstrations for direct imitation, reward shaping, or offline pretraining, Cago treats demonstrations as structured roadmaps. It continuously monitors which parts of a demonstration the agent can already reach and leverages this signal to sample intermediate goal states in the demonstration, those at the boudary of the agent's current goal-reaching capabilities. At each episode, a goal-conditioned agent (Liu et al., 2022; Plappert et al., 2018) first attempts to reach the sampled goal and then explores forward from it, generating informative, task-relevant data for policy optimization. This iterative process of capability-aware goal selection and curriculum-aligned exploration enables steadily progress toward solving the full task.

We evaluate Cago across several sparse-reward environments and demonstrate substantial improvements in both sample efficiency and final task performance over existing imitation-based baselines. Our experiments highlight that capability-aware goal sampling provides a powerful signal for structuring learning, particularly in long-horizon tasks.

## 2 Background and Problem Setup

**Reinforcement learning** (RL) aims to enable agents to learn optimal behaviors through trial-and-error interactions with an environment. An RL problem is formulated as a Markov Decision Process (MDP), represented as a tuple $(\mathcal{S}, \mathcal{A}, \mathcal{T}, \mathcal{G}, \eta, R, \rho_0)$. The agent operates within a state space $\mathcal{S}$ and takes actions from an action space $\mathcal{A}$, transitioning between states according to the dynamics $\mathcal{T}(s'|s, a)$. $R(s, a) \in \mathbb{R}$ is the reward function and $\rho_0$ is the initial state distribution. Given a policy $\pi$, consider the trajectory $\tau = \{s_0, a_0, s_1, a_1, \ldots\}$ sampled by $\pi$, i.e., $s_0 \sim \rho_0$, $a_t \sim \pi(\cdot|s_t)$, and $s_{t+1} \sim \mathcal{T}(\cdot|s_t, a_t)$. The goal of RL is to learn a return-maximizing policy $\pi^* = \arg\max_\pi \mathbb{E}_{\tau \sim \pi(a_t|s_t)} \left[\sum_{t=0}^{\infty} \gamma^t r(s_t, a_t)\right]$ where $\gamma \in (0, 1]$ is the discount factor.

**Learning from Demonstrations.** In imitation learning, the agent is provided a dataset of demonstrations $\mathcal{D}_{\text{demo}}$ collected from some expert policy $\pi_{\text{expert}}$. The objective is to learn a policy $\pi$ that reproduces the expert's behavior by generalizing from these demonstrations. The simplest approach, behavioral cloning (BC), treats this as a supervised learning problem, minimizing the discrepancy between the agent's predicted actions and the expert's. Another line of work, inverse reinforcement learning (IRL) (Abbeel and Ng, 2004), aims to infer an underlying reward function that explains the expert's behavior and then optimizes a policy through RL on this learned reward, thus decoupling reward inference from policy optimization. Building on ideas from IRL, Generative Adversarial Imitation Learning (GAIL) (Ho and Ermon, 2016) bypasses explicit reward recovery by training a policy and a discriminator in an adversarial game: the discriminator distinguishes expert from agent trajectories, while its output serves as an implicit, learned reward signal guiding the policy. More broadly, many recent imitation learning algorithms can be interpreted as minimizing a divergence between the expert and agent occupancy measures.

**State Reset.** Several methods attempt to mitigate the exploration challenge in sparse-reward RL environments by resetting the agent to states from expert demonstrations, thereby bypassing the need to discover those states through the agent's own exploration. These strategies include initializing the agent to states sampled uniformly from demonstration trajectories (Nair et al., 2018; Peng et al., 2018; Hosu and Rebedea, 2016), employing a hand-crafted curriculum (Zhu et al., 2018), or using a reverse curriculum that progressively trains the agent from goal or high-reward states backward (Resnick et al., 2018; Salimans and Chen, 2018; Tao et al., 2024). These approaches assume the ability to reset the agent to arbitrary demonstration states—an assumption that is unrealistic in real-world settings.

**Goal-Conditioned RL** (GCRL) extends the standard RL framework by conditioning policies on specific target goals, guiding agents toward desired goals. The MDPs are augmented with

a goal space $\mathcal{G}$ and are associated with states via a mapping $\eta : \mathcal{S} \rightarrow \mathcal{G}$, ensuring that each state corresponds to an achieved goal. In GCRL, the reward signal from the environment is typically sparse and is defined as: $R(s, a, s', g) = 1\{\eta(s') = g\}$. We assume that each episode has a fixed horizon $T$ and $\mathcal{S} = \mathcal{G}$. The agent's objective is to train a goal-conditioned policy $\pi^G(\cdot | s_t, g)$ to achieve a given goal $g \in \mathcal{G}$ through maximizing the expected cumulative reward $J(\pi) = \mathbb{E}_{g \sim p_g, \tau \sim \pi^G(a_t | s_t, g)} \left[ \sum_{t=0}^{T-1} \gamma^t \cdot R(s_t, a_t, s_{t+1}, g) \right]$ where $p_g$ is the goal distribution.

**This Paper.** We introduce Cago, a novel approach that leverages demonstrations as a scaffold for goal-directed reinforcement learning. Rather than direct imitation, Cago uses demonstrations to guide exploration by training a goal-conditioned policy $\pi^G(a \mid s, g)$ that learns to progressively reach intermediate states $g$ along demonstration trajectories, effectively inducing a curriculum that facilitates steady progress toward solving the full task. In addition, Cago learns a goal predictor $\mathcal{P}(s)$ that infers the final goal state $g_T$ from the current state $s$. The resulting task policy is defined as $\pi(s) = \pi^G(s, \mathcal{P}(s))$ that enables automatic inference of goal conditions at test time for previously unseen situations.

## 3 Method

The main idea of Cago is to continuously monitor the agent's evolving capabilities to reach various stages of demonstration trajectories during training. It dynamically selects the most appropriate goal from the demonstrations, conditioned on the agent's current performance ceiling. The selected goal guides online exploration, with the agent first attempting to reach it using its current policy. From there, it continues to explore, collecting task-relevant trajectories in an Go-Explore style (Ecoffet et al., 2019). By anchoring exploration in achievable yet progressively harder goals, this process effectively constructs an implicit curriculum, where the agent is gradually exposed to more challenging states aligned with its growing competence.

### 3.1 Observation Visit Tracking with Demonstration Alignment

Cago assumes the existence of a limited amount of expert demonstrations $\mathcal{D}_{\text{demo}} : \{\tau^{(i)} = \{(s_0, a_0)^{(i)}, \ldots, (s_{L_i}, a_{L_i})^{(i)}\}\}_{i=1}^M$ where $M$ is the number of the demonstrations and $L_i$ is the length of $i$-th demonstration $\tau^{(i)}$. To select goals at the boundary of the agent's current reaching capabilities, It is crucial to determine the stage at which the agent can accomplish its task completion. Central to our method is maintaining a dictionary $\text{Dict}_{\text{visit}}$ that tracks the visitation frequencies of observations $s_i$ across demonstrations. For each demonstration $\tau^{(i)}$, we initialize an all-zero list of the same length as its steps. Each element in this list records the visitation count of the corresponding observation in the demonstration. At each environment step, the record list is updated to reflect whether the agent has visited observations from the demonstration, based on similarity metrics $\text{sim}(\cdot, \cdot)$ such as L2 distances for state-based environments or mean squared errors (MSE) between images in visual environments. Formally, we define the visitation record dictionary as:

$$\text{Dict}_{\text{visit}} = \tau^{(i)} : [0, 0, \ldots, 0] \in \mathbb{N}^{L_i} \mid i = 1, 2, \ldots, M \tag{1}$$

During online exploration, for each new episode, we first sample a demonstration $\tau^{(i)}$ from $\mathcal{D}_{\text{demo}}$ and reset the environment to the initial state of $\tau^{(i)}$. **Our strategy is more practical in the real-world setting** than related methods (e.g. (Tao et al., 2024; Nair et al., 2018)) that reset the environment to intermediate demonstration states, which are often infeasible to reproduce due to unobservable or difficult-to-control physical factors such as velocity and angular momentum. Given a rollout $\tau = (s_0, s_1, \ldots)$ from the environment, we update $\text{Dict}_{\text{visit}}[\tau^{(i)}]$ as follows:

$$\text{Dict}_{\text{visit}}[\tau^{(i)}][j] \mathrel{+}= 1 \quad \text{if } \text{sim}(s_t, s_j^{(i)}) \leq \epsilon, \quad \forall t \in 1, \ldots, L_\tau, \forall j \in 1, \ldots, L_i \tag{2}$$

where $s_t$ is the agent's observation state at timestep $t$, $L_\tau$ is the total length of the rollout $\tau$, $s_j^{(i)}$ is the $j$-th observation state in the $i$-th demonstration $\tau^{(i)}$, $\text{sim}(\cdot, \cdot)$ is the similarity metric (e.g., L2 distance for state-based environments or MSE for image-based environments), and $\epsilon$ is a matching threshold. This simple record dictionary effectively tracks the agent's progress and helps identify its goal-reaching capability limits along task demonstrations.

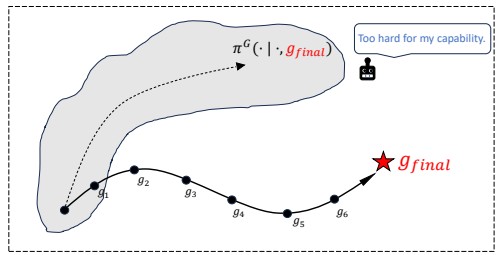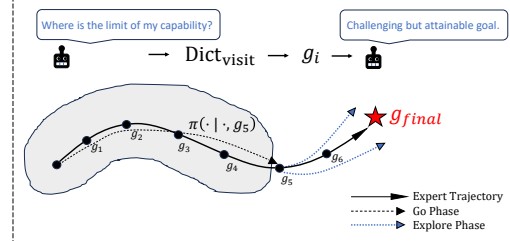

Figure 1: Illustration of the Cago. **Left:** Directly setting the final goal as the agent's target often leads to failure, as the current policy $\pi^G$ may not yet be capable of reaching it. The shaded region illustrates the set of states currently reachable under $\pi^G$. Attempting to reach $g_{\text{final}}$ (i.e., executing $\pi^G(\cdot|\cdot, g_{\text{final}})$) causes the agent to diverge from the demonstration trajectory. **Right:** Cago improves learning by leveraging a visitation frequency dictionary $\text{Dict}_{\text{visit}}$ built from demonstrations. Given a demonstration trajectory with subgoals $g_1, g_2, \ldots, g_n$, the agent selects the furthest subgoal $g_i$ that remains within its current capabilities for Go-Explore sampling, enabling a curriculum of progressively more challenging goals aligned with the demonstration.

## 3.2 Capability-Aware Goal Sampling

Cago leverages demonstration-based visitation counts to guide goal selection and trajectory collection in a *capability-aware* manner. Figure 1 illustrates our method. After resetting the environment to the initial state of a randomly sampled demonstration $\tau^{(i)}$, we select a goal $g$ for the agent to explore:

$$g \sim \mathcal{G}_{\text{cap}}(\pi^G, \tau^{(i)}), \tag{3}$$

where $\mathcal{G}_{cap}(\pi^G, \tau^{(i)})$ denotes a capability-aware goal sampling distribution over subgoals whose reachability is aligned with the current goal-reaching capability of the policy $\pi^G$. Cago examines the visitation frequency list to identify the last index where the frequency exceeds a predefined threshold:

$$j^* = \max\left\{ j \mid \text{Dict}_{\text{visit}}[\tau^{(i)}][j] \geq \lambda_{\text{visit}} \right\}, \tag{4}$$

where $\text{Dict}_{\text{visit}}[\tau^{(i)}][j]$ denotes the visitation frequency of $j$-th observation $s_j$ of $\tau^{(i)}$ under policy $\pi^G$ and $\lambda_{\text{visit}}$ is a frequency threshold (e.g. 100). This index indicates the latest point in the demonstration that the agent is sufficiently competent at reaching—effectively serving as a proxy for the limit of the agent's current goal-

---

**Algorithm 1** Capability-Aware Goal Sampling (Cago)

1: **Input:** Demonstration $\tau^{(i)}$, Visitation record $\text{Dict}_{\text{visit}}$
2: **Output:** capability-aware goal $g$
3: Identify capability-aware upper limit point $g_i$ using visitation threshold $\lambda_{\text{visit}}$ (Eq. (4))
4: Define sampling region $\mathcal{G}_{cap}(\pi^G, \tau^{(i)})$ centered around $g_i$ (Eq. (5))
5: Sample subgoal $g \sim \mathcal{G}_{cap}(\pi^G, \tau^{(i)})$
6: **return** $g$

---

reaching capability. Cago constructs a goal sampling range centered around $j^*$. The sampled goal is drawn from this range, which allows the agent to either revisit familiar goals or attempt slightly more challenging ones that are just beyond its current capability. This capability-aware goal sampling strategy introduces controlled diversity into the training process and encourages progressive learning, while also avoiding excessively difficult goals that could derail training. The corresponding goal sampling region is defined as:

$$\mathcal{G}_{cap}(\pi^G, \tau^{(i)}) = \left\{ s_k \in \tau^{(i)} \mid |k - j^*| \leq \delta \cdot L_i \right\}, \tag{5}$$

where $L_i$ is the length of $\tau^{(i)}$ and $\delta \in (0, 1]$ controls the window size for goal sampling (e.g., 10% of the demonstration length). Goals are then sampled at random from this set. Our capability-aware goal sampling scheme introduces a curriculum-aligned learning signal that progressively guides the agent with steady improvement toward successful task completion. The overall goal sampling process in Cago is described in Algorithm 1.

## 3.3 Learning Framework

**Go-Explore.** Cago trains a goal-conditioned agent following the Go-Explore paradigm (Ecoffet et al., 2019), which divides each episode into two sequential phases: the *Go-phase* and the *Explore-phase*.

In the Go phase, the agent is guided towards a sampled goal state $g$ using the goal-conditioned policy $\pi^G(\cdot|\cdot, g)$, reaching an intermediate state $s_E$. To improve environment exploration beyond the agent's current capabilities, the Explore-phase takes over from $s_E$, where an exploration policy $\pi^E$ is used for the remaining time steps. Since we have access to a limited set of task demonstrations $\mathcal{D}_{\text{demo}}$, we implement $\pi^E$ as a Behavior Clone (BC) Explorer trained on $\mathcal{D}_{\text{demo}}$. The BC Explorer outputs a stochastic action distribution that enables the agent to balance between exploration and imitation. This two-phase strategy ensures that collected trajectories stay anchored near the demonstration distribution in $\mathcal{D}_{\text{demo}}$, while encouraging exploration. We further analyze the impact of the BC Explorer through ablation studies, detailed in Section 4.

As Cago actively resets environments to initial states drawn from the demonstration set $\mathcal{D}_{\text{demo}}$, a key question is how the agent generalizes beyond $\mathcal{D}_{\text{demo}}$. Our solution is to train the goal-conditioned agent $\pi^G$ using a richer set of imagined rollouts generated by a world model via model-based RL.

**World Model and Policy Training.**
Cago stores the trajectories generated under the Go-Explore paradigm with capability-aware goal sampling in a dataset $\mathcal{D}_{\text{cap}} = \{(s_t, a_t, s_{t+1})_{t=1}^T\}$ for world model and policy training. A predictive world model $\widehat{\mathcal{M}}$ approximates the transition dynamics $\mathcal{T}(s'|s, a)$ in the real world $\mathcal{M}$ as $\widehat{\mathcal{T}}(s'|s, a)$. Our model learning algorithm is based on the Dreamer backbone (Hafner et al., 2019, 2020, 2023), which updates the world model $\widehat{\mathcal{M}}$ via supervised learning using $\mathcal{D}_{\text{cap}}$. Once the world model is updated, we train the goal-conditioned policy $\pi^G$ using imagined trajectories generated by the world model $\widehat{\mathcal{M}}$. Intuitively, since $\mathcal{D}_{\text{cap}}$ is collected by exploring around demon-

---

**Algorithm 2** The main training framework of Cago

1: **Input:** GC Policy $\pi^G$, World Model $\widehat{\mathcal{M}}$, Demonstrations $\mathcal{D}_{\text{demo}}$
2: Initialize replay buffer $\mathcal{D}_{\text{cap}}$ and Dict$_{\text{visit}}$ (Eq. (1))
3: Train explorer policy $\pi^E$ using Behavior Cloning on $\mathcal{D}_{\text{demo}}$
4: Train goal predictor $\mathcal{P}_\phi$ on $\mathcal{D}_{\text{demo}}$ (Eq. (7))
5: **for** $n = 1$ to $N_{train}$ **do**
6:     Initialize empty trajectory $\tau$
7:     Randomly sample a demonstration $\tau^{(i)} \in \mathcal{D}_{\text{demo}}$
8:     Initialize the environment to the initial state of $\tau^{(i)}$
9:     Sample a capability-aware goal $g$ by Algorithm 1
10:     **for** $t = 0$ to $L_\tau$ **do**
11:         **if** agent has not reached $g$ and $t < T_{go}$ **then**
12:             $\pi = \pi^G(s, g)$
13:         **else**
14:             $\pi = \pi^E(s)$
15:         Step in the real environment using $\pi$ and add this step to $\tau$
16:         Update Dict$_{\text{visit}}[\tau^{(i)}]$ (Eq. (2))
17:     $\mathcal{D}_{\text{cap}} \leftarrow \mathcal{D}_{\text{cap}} \cup \{\tau\}$
18:     Update $\widehat{\mathcal{M}}$ with $\mathcal{D}_{\text{cap}}$
19:     Update $\pi^G$ using imagined rollouts with $\widehat{\mathcal{M}}$

---

stration states in $\mathcal{D}_{\text{demo}}$, the learned model enables the agent to generate imagined trajectories that remain grounded in task-relevant regions of the state space. Each imagined trajectory from the learned world model $\widehat{\mathcal{M}}$ begins at $s_0$, a state randomly sampled from a trajectory $\tau$ in $\mathcal{D}_{\text{demo}} \cup \mathcal{D}_{\text{cap}}$, and is rolled out for $H$ steps using the goal-conditioned policy $\pi^G(a_t|s_t, g)$. The goal state $g$ is selected as a future state $s_H$ from the same trajectory $\tau$. The objective is to train $\pi^G$ to reinforce trajectories that efficiently reach $g$ in the imagined rollouts from $s_0$ under the learned dynamics $\widehat{\mathcal{M}}$. To achieve this, we adopt an actor-critic algorithm that leverages a self-supervised temporal distance function $D_t(s, g)$ (Mendonca et al., 2021), which estimates the number of steps required to transition from state $s$ to goal $g$. The reward function is defined as: $r^G(s, g) = -D_t(s, g)$. This formulation encourages the policy to generate actions that minimize the estimated temporal distance to the goal. The temporal distance estimator $D_t$ is trained by extracting state pairs $(s_t, s_{t+k})$ from simulated trajectories generated by the world model. The function learns to predict the normalized temporal difference between two states: $D_t\big(\Psi(s_t), \Psi(s_{t+k})\big) \approx \frac{k}{H}$, where $\Psi$ denotes a transformation applied to states (e.g., embedding them into the world model's latent space), and $H$ is the length of the generated rollout. More details on the model-based learning algorithm and the full training procedure for $D_t$ can be found in Appendix B.1 and Appendix B.2.

**Goal Predictor.** At training time, the goal-conditioned policy $\pi^G(\cdot \mid \cdot, g)$ is trained using intermediate states from demonstration trajectories as goal conditions (recall $\mathcal{S} = \mathcal{G}$). This assumes access to demonstrations, with the final states used as the target goal condition. However, at test time, this assumption no longer holds: for unseen scenarios, the true final goal state is not available. This raises the challenge of how to specify an appropriate goal condition based solely on the agent's current observation. We introduce a goal predictor $\mathcal{P}_\phi$, a learned model

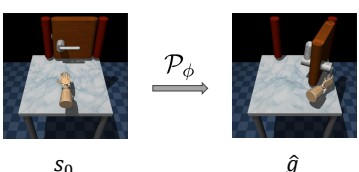

$s_0$          $\hat{g}$

Figure 2: The workflow of the goal predictor $\mathcal{P}_\phi$.

that infers a goal state $\hat{g}$ given the current observation $s$:

$$\mathcal{P}_\phi : s \mapsto \hat{g}, \quad \text{where } \hat{g} = \mathcal{P}_\phi(s) \tag{6}$$

The mapping learned by $\mathcal{P}_\phi$ is illustrated in Figure 2. It is trained using demonstration trajectories $\mathcal{D}_{\text{demo}}$, by minimizing the mean squared error between the predicted goal and the true final observation:

$$\min_\phi \; \mathbb{E}_{(\tau^{(i)} = s_0^{(i)}, \ldots, s_L^{(i)}) \sim \mathcal{D}_{\text{demo}}} \left\| \mathcal{P}_\phi(s_t^{(i)}) - s_L^{(i)} \right\|_2^2 \tag{7}$$

Once trained, the goal predictor enables $\pi^G$ to generalize to new tasks. Given a test-time state $s^{\text{test}}$, the predicted goal $\hat{g}^{\text{test}} = \mathcal{P}_\phi(s^{\text{test}})$ serves as the planning target for the agent $\pi = \pi^G(\cdot \mid s^{\text{test}}, \hat{g}^{\text{test}})$. The complete training pipeline of Cago is detailed in Algorithm 2.

**Rationale Behind Cago's Design.** Let $\mathcal{J}(\pi, \mathcal{M})$ and $\mathcal{J}(\pi^e, \mathcal{M})$ be the expected return of the agent's policy $\pi$ and expert policy $\pi^e$ in the real-world MDP $\mathcal{M}$. We want to bound their return difference:

$$\min_\pi \; |\mathcal{J}(\pi^e, \mathcal{M}) - \mathcal{J}(\pi, \mathcal{M})|, \tag{8}$$

Let $R_{\max}$ be the maximum of the reward with unknown dynamics: $R_{\max} = \max_{(s,a)} \mathcal{R}(s, a)$ and $\rho_\mathcal{M}^\pi(s, a) = (1 - \gamma) \sum_{t=0}^\infty \gamma^t P(s_t = s, a_t = a)$ be the discounted state-action visitation distribution of a policy $\pi$ in the real world MDP $\mathcal{M}$. Suppose that the total variation of learned dynamics model $\widehat{\mathcal{M}}$ from the true transitions $\mathcal{M}$ is bounded such $\mathbb{D}_{\text{TV}}(\mathcal{T}(s, a), \widehat{\mathcal{T}}(s, a)) \leq \alpha \quad \forall (s, a) \in \mathcal{S} \times \mathcal{A}$. According to previous work (Rafailov et al., 2021; DeMoss et al., 2023; Kolev et al., 2024), we have:

$$|\mathcal{J}(\pi^e, \mathcal{M}) - \mathcal{J}(\pi, \mathcal{M})| \leq \underbrace{\alpha \frac{R_{\max}}{(1-\gamma)^2}}_{\text{model prediction error}} + \underbrace{\frac{R_{\max}}{1-\gamma} \mathbb{D}_{\text{TV}}\left( \rho_\mathcal{M}^{\pi^e}, \rho_{\widehat{\mathcal{M}}}^\pi \right)}_{\text{adaptation error}} \tag{9}$$

where $\rho_\mathcal{M}^{\pi^e}$ is the discounted visitation distribution of the expert policy and $\mathbb{D}_{\text{TV}}$ denote total variation distance. The model prediction error with respect to the true environment dynamics can be reduced by collecting more real-world data. In contrast, the adaptation error depends on the total variation distance between the distribution of trajectories generated by policy $\pi$ under the learned world model $\widehat{\mathcal{M}}$ and the expert distribution under the true dynamics $\mathcal{M}$. Thus, the learning problem reduces to bounding the deviation between the behavior of the learned policy $\pi$ under the learned model and the expert behavior under the true environment. To this end, given any $H$-step trajectory $(s_0, s_1, \ldots, s_H)$ sampled from expert demonstrations $\mathcal{D}_{\text{demo}}$, Cago encourages the agent to match expert behavior by rewarding it for reaching the final state $g = s_H$ starting from $s_0$ under the learned dynamics model $\widehat{\mathcal{M}}$ (Line 19 in Algorithm 2).

We further show that Cago effectively reduces the model prediction error by leveraging the BC explore policy $\pi^E = \pi^{\text{BC}}$ for data collection. In the following, we use $d_t^{\mathcal{M}, \pi}$ to denote the marginal state-action distribution at time $t$ induced by policy $\pi$ in the environment $\mathcal{M}$. We assume $d_t^{\mathcal{D}_{\text{demo}}} \approx d_t^{\mathcal{M}, \pi^e}$, where $\mathcal{D}_{\text{demo}}$ is the dataset of demonstrations generated by the expert $\pi^e$, and is sufficiently representative to approximate the true marginal distributions at each timestep. Assuming: (1) $\pi^{\text{BC}}$ accurately approximates the expert policy in $\mathcal{D}_{\text{demo}}$, (2) the world model $\widehat{\mathcal{M}}$ is accurately trained on state transitions induced by $\pi^{\text{BC}}$, and (3) the learned policy $\pi$ generates trajectories in $\widehat{\mathcal{M}}$ that closely match the expert's behavior, we can bound the model prediction error along the imagined rollouts generated by $\pi$ under $\widehat{\mathcal{M}}$:

**Theorem 1.** *Let $\mathcal{M}$ denote the true dynamics model and $\widehat{\mathcal{M}}$ the learned model. Let $\pi_{\text{BC}}$ be a behavior-cloned policy, and $\pi$ a new policy. Let $\mathcal{D}_{demo}$ be a dataset of expert demonstrations from an unknown expert policy. Suppose that, for all $t = 0, 1, \ldots, T$, (1) Closeness of behavior cloning: $\mathbb{D}_{\text{TV}}\left( d_t^{\mathcal{M}, \pi_{\text{BC}}}, d_t^{\mathcal{D}_{demo}} \right) \leq \kappa$, (2) Model learning error under BC: $\mathbb{E}_{(s,a) \sim d_t^{\mathcal{M}, \pi_{\text{BC}}}} \left[ \mathbb{D}_{\text{TV}}\left( \mathcal{M}(\cdot \mid s, a), \widehat{\mathcal{M}}(\cdot \mid s, a) \right) \right] \leq \mu$, and (3) Trajectory distribution closeness: $\mathbb{D}_{\text{TV}}\left( \rho_\mathcal{M}^{\pi^e}, \rho_{\widehat{\mathcal{M}}}^\pi \right) \leq \nu$. Then for all $t = 0, 1, \ldots, T$, we have:*

$$\mathbb{E}_{(s,a) \sim d_t^{\widehat{\mathcal{M}}, \pi}} \left[ \mathbb{D}_{\text{TV}}\left( \mathcal{M}(\cdot \mid s, a), \widehat{\mathcal{M}}(\cdot \mid s, a) \right) \right] \leq \mu + 2\kappa + 2\nu.$$

# 4 Experiments

We evaluate Cago across a diverse set of challenging robotic manipulation environments to address the following research questions: (Q1) Does Cago outperform existing imitation learning baselines that leverage demonstrations in alternative ways? (Q2) Can Cago effectively realize capability-aware goal sampling that aligns with the agent's learning progress? (Q3) How essential are the proposed capability-aware goal sampling and BC-Explorer components to the overall performance of Cago?

**Environments**. For our experiments, we evaluate and compare Cago against several baselines across three robot environment suites with sparse rewards: MetaWorld (Yu et al., 2020), Adroit (Rajeswaran et al., 2017), and Maniskill (Gu et al., 2023; Tao et al., 2025). We adopt the five "very hard" level environments from MetaWorld, as categorized by Seo et al. (2023): Shelf Place, Disassemble, Stick Pull, Stick Push, Pick Place Wall. These environments are considered the most challenging tasks in Metaworld, requiring precise robotic arm control with only sparse task completion rewards. We also use three dexterous hand manipulation tasks from the Adroit suite: Door, Hammer, Pen. To succeed in these three environments, the agent must perform fine-grained and intricate finger manipulations, enabling the grasping and movement of different objects. We also selected three challenging tasks from the ManiSkill benchmark: PegInsertionSide, StackCube, and PullCubeTool. The sparse-reward ManiSkill environments are the most difficult tasks in our benchmarks due to their high-dimensional state and action spaces. During training, we used only 10 demonstration trajectories per task for the MetaWorld and Adroit environments, and 20 demonstration trajectories per task for the ManiSkill environments. More details about each task can be found in Appendix E.

**Baselines**. Our approach is developed on top of the **Dreamer** framework (Hafner et al., 2019, 2020; Hu et al., 2023; Duan et al., 2024a,b), making it a key model-based RL baseline for evaluating the performance gains of Cago. Jump-Start Reinforcement Learning (**JSRL**) (Uchendu et al., 2023) is a curriculum-based approach that leverages a guide-policy pretrained from offline data to guide early-stage exploration during online training. At the beginning of each training episode, the agent follows the guide-policy for a number of steps determined by curriculum progression, after which control is handed over to the online policy. In our JSRL implementation, due to the limited number of demonstrations available, training a reliable guide-policy becomes challenging. Therefore, we directly use the demonstration trajectories as the guide-policy. Specifically, we reset the environment to a demonstration initial state, enabling the agent to replicate expert behavior during the initial phase of each episode before switching to the online policy. Our JSRL implementation is also built on top of the Dreamer framework. **MoDem** (Hansen et al., 2022) represents one of the most efficient frameworks in the model-based RL literature. It pretrains its policy using a small set of demonstrations and repeatedly oversamples the demonstrations to train both the world model and the policy. We consider MoDem to be the strongest baseline due to its fast convergence and low data requirements. **Cal-QL** (Nakamoto et al., 2023) is a state-of-the-art algorithm following the offline-to-online RL paradigm. It uses demonstrations to pretrain the $Q$-function and applies calibration to mitigate performance drop when transitioning from offline to online learning phases. In addition to these representative baselines, we compare against four more imitation learning baselines: **GAIL** (Ho and Ermon, 2016), **PWIL** (Dadashi et al., 2020), **SQIL** (Reddy et al., 2019), **ValueDice** (Kostrikov et al., 2019), and **RLPD** (Ball et al., 2023), a well-tuned variant of SAC that leverages offline data when learning online, in the Appendix F.1.

**Main Results**. During training, we uniformly sample a demonstration and reset the environment using the same seed that was used to collect it. All baselines share the same seeds and demonstration data. To evaluate generalization, we test on unseen environment seeds. For these, Cago uses the goal predictor (Section 3.3) to infer goal conditions for the goal-conditioned policy $\pi^G$. Each method is evaluated on 100 held-out seeds, and we report the average success rate over these episodes. Figure 3 depicts the mean learning performance of Cago and all baselines in terms of the agent's task success rate averaged over 8 random seeds. On the MetaWorld very hard tasks, Cago consistently outperforms all the baselines in both final performance and learning efficiency. In the Adroit suite, although Modem exhibits rapid early learning due to its behavior cloning (BC) pretraining and oversampling strategy, Cago surpasses it in final performance after 1e6 environment interaction steps. Notably, Dreamer, which shares the same world model and policy architecture as Cago, performs significantly worse, underscoring the effectiveness of the capability-aware goal sampling strategy. Our JSRL baseline, based on the same world model architecture, adopts a uniform curriculum to reduce the guide-steps from demonstrations. It lags behind Cago in both learning speed and final success

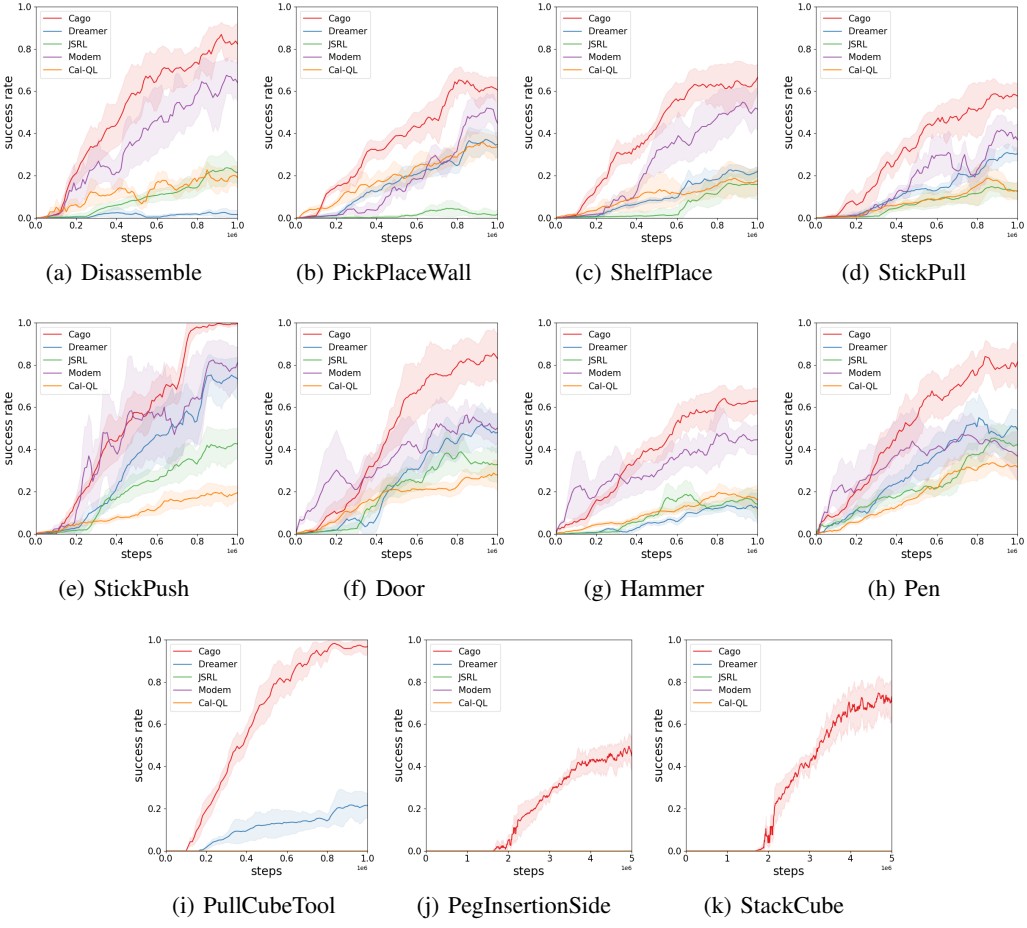

Figure 3: Experiment results comparing Cago with the baselines over 8 random seeds. The solid line denotes the average success rate in *evaluation*, while the shaded region signifies the standard deviation.

rate, highlighting the effectiveness of our goal-sampling strategy in adapting to the agent's evolving capabilities. In the ManiSkill environments, given the limited demonstrations, Cago stands out as the only method capable of achieving high success rates.

**Capability-Aware Goal Distribution**. To answer Q2, we visualize the progression of capability-aware goal sampling throughout the training process in the StickPush environment in Figure 4(d). Each red dot represents the normalized position of a sampled goal within a demonstration trajectory, with 0 indicating the start and 1.0 indicating the final demonstration state. Early in training, the agent predominantly samples goals at lower normalized positions, focusing on subgoals near the beginning of the trajectory that are within its current capabilities. As training advances, goal sampling gradually shifts toward higher normalized positions, indicating the agent's increasing ability to pursue more challenging goals closer to task completion. By continuously targeting goals just at the boundary of the agent's current capability, Cago facilitates efficient learning in sparse-reward, long-horizon tasks.

**Ablation Studies**. To answer Q3, we assess the individual contributions of (a) capability-aware goal sampling and (b) the BC-Explorer component to the overall performance. The first ablation, Cago-FinalGoal, retains only (b): it uses BC-based exploration but always selects the final observation from a demonstration in the goal phase of our Go-Explore sampling paradigm, ignoring the agent's current goal-reaching capability. The BC Explorer takes control from the goal-conditioned policy halfway through each rollout. The second ablation, Cago-StepBased, also retains (b), but samples goals from demonstrations in proportion to training steps (current training step / predefined total training steps). However, it does not assess the agent's actual capabilities, and may therefore sample goals that are either too easy or too difficult for the agent at its current learning stage. The third ablation,

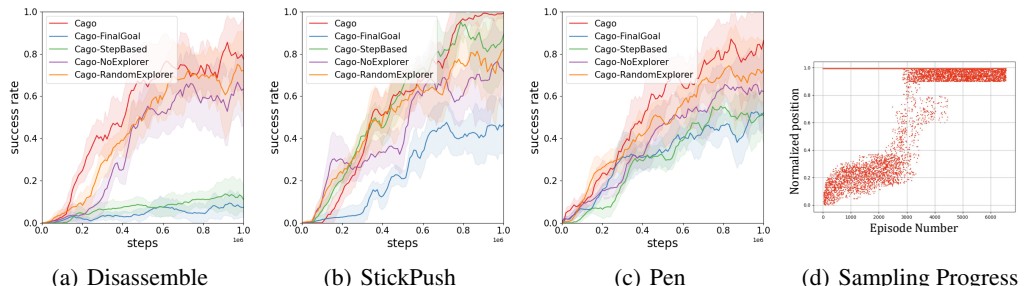

(a) Disassemble     (b) StickPush     (c) Pen     (d) Sampling Progress

Figure 4: Figure(a),(b),(c) are the results of ablation study on the importance of each component of Cago over 5 seeds. Figure(d) shows the progress of capability-aware goal sampling in Stickpush.

Cago-NoExplorer, keeps only (a): it uses capability-aware goal sampling, but does not explore beyond the sampled goal with the BC Explorer. The fourth ablation, Cago-RandomExplorer, replaces the BC Explorer with a uniformly random policy during the exploration phase of our Go-Explore-style rollout strategy. We conduct the ablation study on the Disassemble and StickPush tasks from MetaWorld, and the Pen task from Adroit. As shown in Figure 4, removing capability-aware goal sampling significantly degrades performance. Without it, the agent often enters the Explore phase from states far outside the demonstration region, making it difficult for the BC-Explorer to make meaningful progress. The BC-Explorer itself is also crucial, as it accelerates learning by generating high-quality exploratory rollouts. We further examine how the number and quality of demonstrations (including suboptimal ones), as well as the hyperparameter $\lambda_{visit}$ and $\delta$, affect performance; see Appendix F for details.

**Visual-input Environments**. We further assess Cago's applicability in high-dimensional visual settings. Specifically, we extend our framework to raw pixel observations by replacing the vector-based states and goals with RGB image inputs of size (64, 64, 3), resulting in a variant referred to as Cago-Visual. In this setting, both the policy and the goal predictor $\mathcal{P}_\phi$ operate on image representations. We benchmark Cago-Visual against Modem-Visual, a strong model based baseline that similarly utilizes image-based observations and demonstration. As shown in Figure 5, Cago-Visual not only retains performance similar to the original Cago, but also consistently outperforms Modem-Visual, highlighting the robustness of our method in visual domains. Details of the goal predictor employed in our visual-input experiments are provided in Appendix F.2.

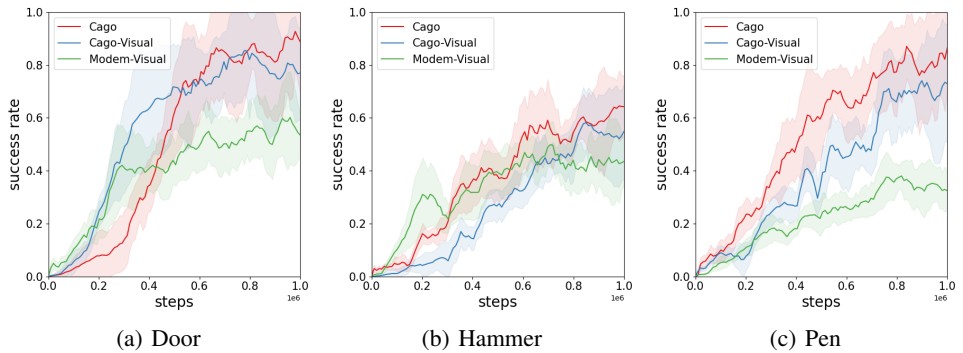

(a) Door       (b) Hammer       (c) Pen

Figure 5: Visual input experiment results over 5 random seeds.

## 5 Related Work

**Imitation Learning.** Demonstrations are a key tool for improving the efficiency of RL, with prior work integrating them across various stages of the RL pipeline (Arulkumaran and Lillrank, 2024; Nair et al., 2018; Cui et al., 2025). A prominent approach uses demonstrations for direct learning via Behavior Cloning (BC) and its variants (Bain and Sammut, 1995; Torabi et al., 2018). The introduction of the Generative Adversarial Imitation Learning (GAIL) algorithm (Ho and Ermon,

2016) has driven significant advances in scalable deep imitation learning methods (Fu et al., 2017; Ghasemipour et al., 2020; Kostrikov et al., 2018; Jena et al., 2020; Finn et al., 2016; Blondé and Kalousis, 2019; Orsini et al., 2021; Eysenbach et al., 2021). Beyond adversarial approaches, several imitation learning algorithms aim to match the state action distributions of the expert and the agent through non-adversarial techniques, such as non-parametric models (Kim and Park, 2018), random network distillation (Wang et al., 2019), support estimation (Brantley et al., 2020), Wasserstein distance minimization (Dadashi et al., 2020), and moment matching (Swamy et al., 2021). Demonstrations have also been the bridge between imitation learning and offline RL. Conservative Q-learning (CQL) (Kumar et al., 2020) and Cal-QL (Nakamoto et al., 2023) regularize $Q$-values using demonstration data to better estimate out-of-distribution actions. Traditional imitation learning typically requires direct learning from demonstrations, either by recovering a reward function from demonstrations, or matching expert and agent state-action distributions (e.g. GAIL(Ho and Ermon, 2016)), or incorporating demonstrations as anchors to mitigate overestimation and instability caused by out-of-distribution actions (e.g. Cal-QL(Nakamoto et al., 2023)). In contrast, Cago introduces a novel use of demonstrations by treating them as a structured roadmap for building an adaptive curriculum, scaffolding the agent's learning to enable steady progress toward solving the full task.

**Curriculum for Learning from Demonstrations.** Several prior works have explored curriculum design using demonstration. Yengera et al. (2021); Tao et al. (2024) introduce difficulty scores to rank demonstrations, offering a theoretical framework for selecting optimal trajectories to scaffold learning. Task Phasing (Bajaj et al., 2023) automatically extracts curriculum phases from demonstrations and dynamically transitions the agent through them during training. JSRL (Uchendu et al., 2023) design a curriculum-based approach that leverages a guide-policy pretrained from demonstration to guide early-stage exploration during online training. Compared to these curriculum learning methods using demonstration, Cago employs a *goal-level* curriculum that incrementally samples intermediate goals from demonstrations based on the agent's evolving capabilities. A number of methods have explored the use of expert demonstration states as informative priors for guiding agent learning. A common strategy involves resetting the agent from states sampled along demonstration trajectories (Nair et al., 2018; Peng et al., 2018; Hosu and Rebedea, 2016), enabling the agent to experience these expert regions of the state space without exhaustive exploration. Some approaches employ structured curricula to sequence these resets, either through manually designed progressions (Zhu et al., 2018) or automated strategies such as reverse curricula, which gradually increase the exploration horizon by training the agent from goal states backward (Resnick et al., 2018; Salimans and Chen, 2018; Tao et al., 2024). While effective in simulation, these methods rely on the ability to precisely reset the environment to arbitrary demonstration states—a requirement that poses significant challenges in physical systems, where replicating exact configurations, including latent dynamics like joint velocities, is often infeasible. Rather than forcibly placing the agent into expert states—a strong form of intervention—Cago encourages the agent to actively reach intermediate goals that are sampled to match its current competence. This self-directed learning process allows the agent to internalize problem-solving skills more effectively, promoting deeper understanding through its own attempts rather than directly resetting. While demonstrations provide useful guidance, true mastery requires the agent to explore and overcome challenges through trial and error.

# 6   Conclusion

We introduce Cago, a novel method that leverages demonstrations in a dynamic, goal-guided manner to tackle exploration challenges in sparse-reward environments. By continuously monitoring the agent's capabilities, Cago constructs an adaptive curriculum that incrementally samples intermediate goals from demonstrations, effectively scaffolding learning and enabling steady progress toward solving the full task. Extensive experiments show consistent improvements over baselines. A key limitation of Cago lies in its reliance on resetting environments to the initial states specified in the demonstrations. Appendix C provides a discussion on how this requirement could be relaxed to improve applicability in real-world settings.

# Reproducibility Statement

The code for Cago is available at `https://github.com/RU-Automated-Reasoning-Group/Cago`. For hyperparameter settings, please refer to Appendix H.

## Acknowledgements

We thank the anonymous reviewers for their comments and suggestions. This work was supported by NSF Award #CCF-2124155.

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

# Appendix

## A  Proof of Theorem 1

**Theorem 1**. Let $\mathcal{M}$ denote the true dynamics model and $\widehat{\mathcal{M}}$ the learned model. Let $\pi_{\text{BC}}$ be a behavior-cloned policy, and $\pi$ a new policy. Let $\mathcal{D}_{\text{demo}}$ be a dataset of expert demonstrations from an unknown expert policy. Suppose that, for all $t = 0, 1, \ldots, T$, (1) Closeness of behavior cloning: $\mathbb{D}_{TV}\left(d_t^{\mathcal{M}, \pi_{\text{BC}}}, d_t^{\mathcal{D}_{\text{demo}}}\right) \leq \kappa$, (2) Model learning error under BC: $\mathbb{E}_{(s,a) \sim d_t^{\mathcal{M}, \pi_{\text{BC}}}}\left[\mathbb{D}_{TV}\left(\mathcal{M}(\cdot \mid s, a), \widehat{\mathcal{M}}(\cdot \mid s, a)\right)\right] \leq \mu$, and (3) Trajectory distribution closeness: $\mathbb{D}_{TV}\left(\rho_{\mathcal{M}}^{\pi^e}, \rho_{\widehat{\mathcal{M}}}^{\pi}\right) \leq \nu$. Then for all $t = 0, 1, \ldots, T$, we have:

$$\mathbb{E}_{(s,a) \sim d_t^{\widehat{\mathcal{M}}, \pi}}\left[\mathbb{D}_{TV}\left(\mathcal{M}(\cdot \mid s, a), \widehat{\mathcal{M}}(\cdot \mid s, a)\right)\right] \leq \mu + 2\kappa + 2\nu.$$

*Proof.* By triangle inequality on expectations and total variation:

$$\mathbb{E}_{(s,a) \sim d_t^{\widehat{\mathcal{M}}, \pi}}\left[\mathbb{D}_{\text{TV}}\left(\mathcal{M}(\cdot \mid s, a), \widehat{\mathcal{M}}(\cdot \mid s, a)\right)\right]$$
$$\leq \mathbb{E}_{(s,a) \sim d_t^{\mathcal{D}_{\text{demo}}}}\left[\mathbb{D}_{\text{TV}}\left(\mathcal{M}(\cdot \mid s, a), \widehat{\mathcal{M}}(\cdot \mid s, a)\right)\right] + 2\,\mathbb{D}_{\text{TV}}\left(d_t^{\widehat{\mathcal{M}}, \pi}, d_t^{\mathcal{D}_{\text{demo}}}\right).$$

We now bound each term separately:

- For the first term, apply triangle inequality again:

$$\mathbb{E}_{(s,a) \sim d_t^{\mathcal{D}_{\text{demo}}}}\left[\mathbb{D}_{\text{TV}}\left(\mathcal{M}(\cdot \mid s, a), \widehat{\mathcal{M}}(\cdot \mid s, a)\right)\right] \leq \mathbb{E}_{(s,a) \sim d_t^{\mathcal{M}, \pi_{\text{BC}}}}\left[\mathbb{D}_{\text{TV}}\left(\mathcal{M}(\cdot \mid s, a), \widehat{\mathcal{M}}(\cdot \mid s, a)\right)\right]$$
$$+ 2\,\mathbb{D}_{\text{TV}}\left(d_t^{\mathcal{D}_{\text{demo}}}, d_t^{\mathcal{M}, \pi_{\text{BC}}}\right)$$
$$\leq \mu + 2\kappa,$$

  using assumptions (1) and (2).

- For the second term, use that marginal total variation is bounded by trajectory total variation:

$$\mathbb{D}_{\text{TV}}\left(d_t^{\widehat{\mathcal{M}}, \pi}, d_t^{\mathcal{D}_{\text{demo}}}\right) \leq \mathbb{D}_{\text{TV}}\left(\rho^{\widehat{\mathcal{M}}, \pi}, \rho^{\mathcal{M}, \pi^e}\right) \leq \nu,$$

  by assumption (3).

Combining:

$$\mathbb{E}_{(s,a) \sim d_t^{\widehat{\mathcal{M}}, \pi}}\left[\mathbb{D}_{\text{TV}}\left(\mathcal{M}(\cdot \mid s, a), \widehat{\mathcal{M}}(\cdot \mid s, a)\right)\right] \leq \mu + 2\kappa + 2\nu.$$

$\square$

## B  Extended Background

### B.1  Dreamer World Model

The RSSM consists of an encoder, a recurrent model, a representation model, a transition predictor, and a decoder, as formulated in Equation 10. And it employs an end-to-end training methodology, where its parameters are jointly optimized based on the loss functions of various components, including dynamic transition prediction, reward prediction, and observation encoding-decoding. These components often operate in a latent space rather than the original observation space, as encoded by the World Model. Therefore, during end-to-end training, the losses of all components indirectly optimize the latent space.

The encoder $f_E$ encodes the input state $x_t$ into a embed state $e_t$, which is then fed with the deterministic state $h_t$ into the representation model $q_\varphi$ to generate the posterior state $z_t$. The transition

predictor $p_\varphi$ predicts the prior state $\hat{z}_t$ based on the deterministic state $h_t$ without access to the current input state $x_t$. Using the concatenation of either $(h_t, z_t)$ or $(h_t, \hat{z}_t)$ as input, the recurrent transition function $f\varphi$ iteratively updates the deterministic state $h_t$ with given action $a_t$.

$$
\begin{aligned}
\text{Encoder:} \quad & e_t = f_E(e_t|x_t) \\
\text{Recurrent model:} \quad & h_t = f_\varphi(h_{t-1}, z_{t-1}, a_{t-1}) \\
\text{Representation model:} \quad & z_t \sim q_\varphi(z_t|h_t, e_t) \\
\text{Transition predictor:} \quad & \hat{z}_t \sim p_\varphi(\hat{z}_t|h_t) \\
\text{Decoder:} \quad & \hat{x}_t \sim f_D(\hat{x}_t|h_t, z_t)
\end{aligned}
\tag{10}
$$

### B.2 Temporal Distance Training in LEXA

The goal-reaching reward $r^G$ is defined by the self-supervised temporal distance objective (Mendonca et al., 2021) which aims to minimize the number of action steps needed to transition from the current state to a goal state within imagined rollouts. We use $b_t$ to denote the concatenate of the deterministic state $h_t$ and the posterior state $z_t$ at time step $t$.

$$
b_t = (h_t, z_t)
\tag{11}
$$

The temporal distance $D_t$ is trained by sampling pairs of imagined states $b_t, b_{t+k}$ from imagined rollouts and predicting the action steps number $k$ between the embedding of them, with a predicted embedding $\hat{e}_t$ from $b_t$ to approximate the true embedding $e_t$ of the observation $x_t$.

$$
\text{Predicted embedding:} \quad emb(b_t) = \hat{e}_t \approx e_t, \quad \text{where} \quad e_t = f_E(x_t)
\tag{12}
$$

$$
\text{Temporal distance:} \quad D_t(\hat{e}_t, \hat{e}_{t+k}) \approx k/H \quad \text{where} \quad \hat{e}_t = emb(b_t) \quad \hat{e}_{t+k} = emb(b_{t+k})
\tag{13}
$$

$$
r_t^G(b_t, b_{t+k}) = -D_t(\hat{e}_t, \hat{e}_{t+k})
\tag{14}
$$

## C  Discussion

In this section, we discuss several key questions and extensions regarding the design, applicability, and limitations of Cago. Our goal is to clarify how Cago fundamentally differs from existing imitation learning paradigms, what design choices drive its effectiveness, and how it can be adapted or extended to broader settings. We also explain and clarify its current assumptions—such as initial state resetting and model-based training. Through this discussion, we aim to provide deeper insights into the generality, scalability, and future potential of Cago as a new framework for learning from demonstrations.

### 1. How does Cago differ from traditional Imitation Learning methods?

Cago is fundamentally different from traditional Imitation Learning (IL) methods, both in methodology and motivation. IL typically requires direct learning from demonstrations, either by recovering a reward function from demonstrations, or matching expert and agent state-action distributions (e.g. GAIL(Ho and Ermon, 2016)), or incorporating demonstrations as anchors to mitigate overestimation and instability caused by out-of-distribution actions (e.g. Cal-QL(Nakamoto et al., 2023)). In contrast, Cago presents a new paradigm to utilize demonstrations: it dynamically tracks the agent's competence along demonstrated trajectories and uses this signal to select intermediate states in the demonstrations that are just beyond the agent's current reach as goals to guide online trajectories collecting. To evaluate this novel perspective, we compared Cago against state-of-the-art baselines that represent diverse strategies for learning from demonstrations, including distribution mathcing (e.g. GAIL(Ho and Ermon, 2016) and PWIL(Dadashi et al., 2020)), curriculum learning (e.g. JSRL(Uchendu et al., 2023)), model-based exploration (e.g. MoDem(Hansen et al., 2022)), and offline-to-online fine-tuning (e.g. CalQL(Nakamoto et al., 2023) and RLPD(Ball et al., 2023)).

**2. Can Cago be extended with more sophisticated similarity metrics to handle complex tasks?**

In our implementation of Cago, we use the observation space as the goal space and adopt simple similarity metrics such as MSE or L2 distance to demonstrate the applicability and robustness of Cago's curriculum strategy in both state-based and pixel-based environments. Cago can be naturally extended to support more complex settings. For example, in high-dimensional visual environments, Cago could adopt more expressive similarity metrics such as SSIM (Structural Similarity Index) or FSIM (Feature Similarity Index). In tasks involving semantic or language-conditioned goals, one could compute a scalar score function that estimates how well a visual state satisfies a goal, and then compare states based on the difference in their scores with respect to the same goal. This score function can be instantiated using pretrained vision-language models, allowing goal representations to move beyond raw states to language embeddings or other semantic forms. Our main contribution is a general framework for goal sampling from demonstrations—based on the agent's capability—which can be paired with any goal space and similarity metric appropriate to the domain.

**3. How does Cago perform relative to curriculum learning baselines?**

In our experiments, we included Jump-Start Reinforcement Learning (JSRL(Uchendu et al., 2023)), a recent and strong curriculum learning baseline. JSRL pretrains a guide policy on offline data to provide a curriculum of starting states for the RL policy, which significantly simplifies the exploration problem. As the RL policy improves, the effect of the guide-policy is gradually diminished, leading to an RL-only policy that is capable of further autonomous improvement. We discuss four additional curriculum learning baselines: Yengera et al. (2021); Bajaj et al. (2023); Dai et al. (2021); Hermann et al. (2020). Yengera et al. (2021) constructs a personalized demonstration curriculum by computing task-specific difficulty scores based on the teacher's and learner's policies. In contrast, Cago does not require hand-crafted difficulty metrics—instead, it extracts intermediate goals directly from the demonstration, and guides the agent to reach them progressively based on its own learning progress. This avoids the challenge of designing difficulty metrics for complex manipulation tasks, which is non-trivial without expert knowledge. Bajaj et al. (2023) assumes access to a demonstrator policy, either via imitation learning or inverse reinforcement learning. However, our experiments show that standard imitation learning performs poorly on our benchmarks, limiting the effectiveness of such approaches in our domain. Dai et al. (2021); Hermann et al. (2020) segment demonstrations and construct a curriculum by resetting episodes to intermediate states along the demonstration trajectory, starting from the end and moving backward as learning progresses. Cago explicitly avoids such state resets, which are often unrealistic in real-world robotic settings without simulator support. we conducted additional experiments comparing Cago to **ACED**(Dai et al., 2021) and **Task phasing**(Bajaj et al., 2023) on the Fetch-PickPlace and Fetch-Slide environments that they were evaluated on. The tables 1 below report the final success rates after 1M training steps, averaged over 5 random seeds (We were unable to run ACED on Fetch-Slide).

Table 1: Comparison of Cago and ACED, Task phasing across Fetch-PickPlace and Fetch-Slide tasks.

| Task | Cago (Ours) | ACED (Dai et al., 2021) | Task phasing (Bajaj et al., 2023) |
|---|---|---|---|
| Fetch-PickPlace | 1.00 | 0.93 | 0.75 |
| Fetch-Slide | 0.48 | – | 0.20 |

**4. Is the need for environment resets in Cago a fundamental limitation?**

Indeed, resetting to specific initial states is a limitation of Cago, but as we mentioned in the Section 5, it is already a substantial improvement over arbitrary state resets, which are often infeasible in real-world robotics. Importantly, Cago does not require exact resets to the full demonstration initial states. For instance, in the Peg Insertion task, the peg's initial position can differ from the demonstration, and Cago can learn to reach states within the early portion of the demonstration. However, some elements—like the hole and the box—are non-movable and randomly initialized by the environment. Since the agent cannot manipulate these objects, we have to reset them to match the demonstration configuration to ensure goal reachability. The constraint over other controllable components, such as the robot arm and peg, can be relaxed. This partial reset strategy reduces the burden and makes Cago more practical for real-world settings.

### 5. What motivates the choice of a model-based RL formulation for Cago, rather than a model-free alternative?

In early experiments, we attempted to train Cago in a model-free manner by learning the goal-conditioned policy directly from real environment rollouts. But we further find that instead of directly using $\mathcal{D}_{cap}$ to train $\pi^G$ in Model-free manner, the world model can offer a richer set of imagined rollouts to train the $\pi^G$, which improves the learning efficiency. In our setting, the simulated trajectories have starting states and goal states randomly sampled from the same trajectory in $\mathcal{D}_{cap}$. As a result, the simulated trajectories still resemble segments from $\mathcal{D}_{cap}$, while significantly increasing trajectory diversity and data richness. The MBRL—leveraging simulated trajectories allows us to generate more diverse and abundant training data. This promotes generalization across the environment. Cago builds upon the **Dreamer** framework (Hafner et al., 2019, 2020; Hu et al., 2023; Duan et al., 2024a,b), which serves as the model-based cornerstone of our approach. Dreamer learns a latent world model and trains a goal-conditioned policy using an actor-critic algorithm. It samples trajectories by setting the final observation of a demonstration as the goal, while resetting the environment using the same seed as the corresponding demonstration. **PEG** (Hu et al., 2023) introduces an alternative goal-selection strategy that enhances exploration within the Dreamer framework. To demonstrate the improvement brought by Cago over its cornerstone, we conduct comparisons against both Dreamer and PEG. Table 2 presents the performance comparison of Cago against Dreamer and PEG across three environments(results are averaged over 5 random seeds). Cago clearly surpasses its model-based cornerstone, Dreamer, as well as the competitive goal-picking strategy, PEG, across all environments.

Table 2: Performance comparison of Cago against Dreamer and PEG.

| Environment | Cago (Ours) | Dreamer | PEG |
|---|---|---|---|
| StickPush | 0.99 | 0.68 | 0.76 |
| Disassemble | 0.80 | 0.02 | 0.04 |
| Adroit-Pen | 0.82 | 0.52 | 0.54 |

### 6. Can Cago be applied to broader scenarios—such as real-world settings with diverse initial states and uncertain dynamics?

Cago can be extended to support diverse initial states and uncertain dynamics. For diverse initial states in real-world settings, Cago can gradually prioritize those where the agent has the most learning potential—starting from demonstration initial states and expanding to nearby ones where the agent can still benefit from the demonstration trajectories for dynamic capability tracking and progressive goal selection. As training progresses, we incorporate successful rollouts from these nearby initial states into the demonstration set, broadening its coverage. This enables a natural curriculum that eventually transitions to diverse, real-world initial states. For uncertain, stochastic dynamics, Cago remains robust by not requiring exact reproduction of actions in demonstration trajectories. Instead, it adapts goals based on observed agent successes, making the curriculum self-correcting in the face of noise or variability. If the agent consistently fails to reach a specific demonstration state, Cago's capability metric naturally shifts goal sampling toward more achievable targets. Additionally, goal selection can be extended with probabilistic estimates—such as goal-reaching likelihood or future state predication uncertainty from an ensemble of world models—which we view as a promising direction for high-stochasticity domains.

### 7. Does Cago risk overfitting to a limited number of demonstrations?

Cago mitigates overfitting to a small number of demonstrations by training its goal-conditioned policy $\pi^G$ within a world model. Rather than training $\pi^G$ to reach states solely from demonstrations, Cago instructs $\pi^G$ to learn to reach diverse states sampled from real rollouts in $\mathcal{D}_{cap}$, which records the trajectories generated under our Go-Explore paradigm with capability-aware goal sampling, in its world model. This promotes generalization across the environment. To evaluate the generalization capability, we tested Cago on 500 unseen initial states generated from random seeds, each differing from those in the demonstrations. We report both the average L2 deviation from demonstration initial states and the average success rate across 500 unseen seeds across three environments in Table 3. The results are consistent with Figure 3.

Table 3: Initial demo observation deviation and average success rate across 500 unseen seeds.

| Environment | Average Initial Observation Deviation | Average Success Rate(500 seeds) |
|---|---|---|
| StickPush | 0.0547 | 0.982 |
| Disassemble | 0.1405 | 0.806 |
| Adroit-Pen | 1.2749 | 0.824 |

# D  Limitations and Future Work

While Cago shows promising results in leveraging demonstrations to improve exploration and learning efficiency in sparse-reward, long-horizon tasks, several limitations remain. In our experiments, we employ image MSE similarity(or L2 distance for state) and a simple counting scheme, which are generalizable and applicable across diverse environments, as illustrative measures of evaluating agent capability in Cago. However, in much more complex environments, such simple approaches may not constitute the optimal strategy. In the future, incorporating uncertainty-aware models for both agent capability estimation and subgoal selection could improve robustness and adaptability of Cago framework. Extending Cago to multi-agent or real-world robotics scenarios is another promising direction, where the complexity of coordination and physical constraints introduces new challenges for efficient demonstration utilization and goal guidance.

# E  Environment Details

We evaluate and compare Cago against several baselines across three robot environment suites using sparse rewards: MetaWorld (Yu et al., 2020), Adroit (Rajeswaran et al., 2017), and Maniskill (Gu et al., 2023; Tao et al., 2025). In this section, we provide more details about each benchmark and the specific experimental setup.

## E.1  MetaWorld

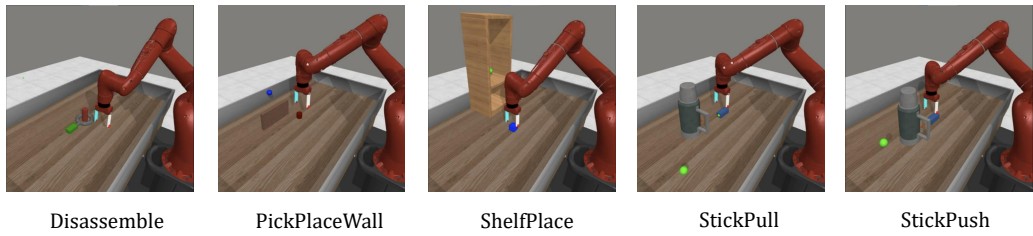

| Disassemble | PickPlaceWall | ShelfPlace | StickPull | StickPush |

Figure 6: 5 very hard level environments from MetaWorld

MetaWorld (Yu et al., 2020) is a widely used benchmark suite designed to evaluate the generalization and manipulation capabilities of reinforcement learning algorithms in robotic control tasks. It consists of a diverse collection of continuous control environments simulated using the MuJoCo physics engine. Each task requires a robotic arm to interact with objects in the scene to achieve goal-directed behavior under sparse or dense reward settings. MetaWorld includes 50 distinct tasks of varying difficulty, ranging from simple reaching to complex object manipulation. In our experiments, we focus on the "very hard" subset of tasks identified in prior studies (Seo et al., 2023; Hansen et al., 2022), which are characterized by sparse rewards, delayed feedback, and the need for precise low-level control, making them particularly suitable for benchmarking sample efficiency and generalization in demonstration-augmented learning frameworks. The five very hard tasks we choose are: Shelf Place, Disassemble, StickPull, Stick Push, Pick Place Wall. Shelf Place: The agent must grasp an object and place it accurately onto a shelf, requiring precise vertical and lateral arm control. Disassemble: The task involves picking a nut out of the a peg, demanding a strong grasp and directional pulling motion. Stick Pull: The robot needs to grasp a stick and pull a bottle, requiring fine force control and coordinated motion. Stick Push: The goal is to grasp a stick and push a bottle, emphasizing

controlled contact and alignment with the target location. Pick Place Wall: The agent must pick up an object and place it over a wall barrier onto a specified target location, combining lifting, positioning, and obstacle avoidance. We use the L2-distance to calculate the similarity between observations to judge if the agent has reached the demonstration observation. The threshold for similarity is 0.05. We use 10 demonstrations for training.

## E.2 Adroit

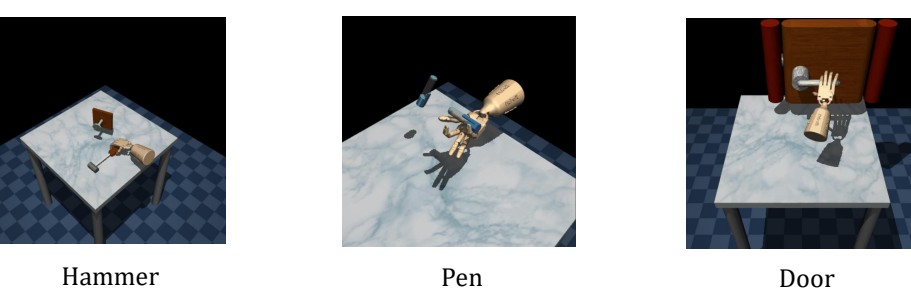

| Hammer | Pen | Door |

Figure 7: 3 environments from Adroit Suite

The Adroit Suite (Rajeswaran et al., 2017) is a set of high-dimensional dexterous hand manipulation tasks that emphasize fine motor control, contact-rich interactions, and sparse-reward learning. It is built upon a 24-DoF ShadowHand robotic hand, presenting significant challenges in both control and generalization. Each task requires the agent to coordinate multiple fingers and joints to manipulate objects with high precision under partial observability and complex dynamics. We use three environments from this suit. Hammer: The agent must grasp a hammer and use it to drive a nail into a box. This task demands stable object manipulation, precise tool orientation, and effective force transmission. Pen: The objective is to reorient and position a pen in an assigned direction. It requires careful control of finger articulation and rotational dexterity. Door: The task involves unlatching and opening a door by manipulating the handle and applying a pulling motion. It tests the agent's ability to perform multi-stage interactions and coordinate wrist and finger movement to exert torque in the correct direction. We use the Mean square error between images rendered to calculate the similarity to judge if the agent has reached the demonstration observation. Each rendered image will be reshaped to a size (100,100,3). We use 10 demonstrations for training.

## E.3 Maniskill

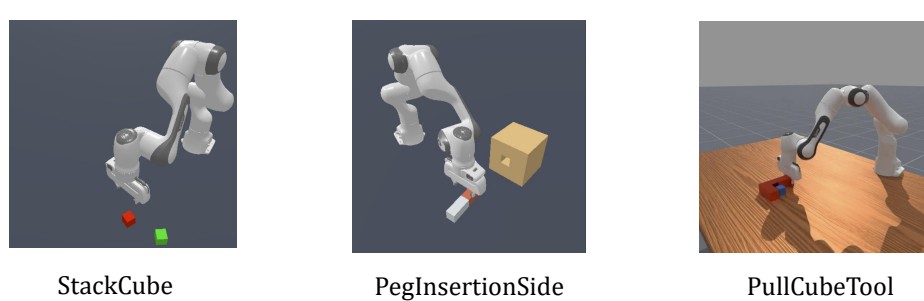

| StackCube | PegInsertionSide | PullCubeTool |

Figure 8: 3 environments from Maniskill

ManiSkill (Gu et al., 2023; Tao et al., 2025) is a comprehensive benchmark suite designed to evaluate generalizable robotic manipulation skills in simulation, emphasizing real-world task diversity, object variety, and generalization across instances. It provides high-quality 3D environments with continuous control, supporting both visual and proprioceptive observations. The benchmark is particularly challenging under sparse reward settings, as tasks often require multi-step reasoning, long-horizon

planning, and precise control to accomplish. We pick three environments from this benchmark. PullCubeTool: Given an L-shaped tool that is within the reach of the robot, the agent needs to leverage the tool to pull a cube that is out of it's reach. PegInsertionSide: The robot must align and insert a peg into a side-entry slot. The task demands precise pose estimation, spatial reasoning, and careful control to avoid misalignment or jamming. StackCube: This task involves picking up a cube and accurately stacking it on top of another. We use the Mean square error between images rendered to calculate the similarity to judge if the agent has reached the demonstration observation. Each rendered image will also be reshaped to a size (100,100,3). We use 20 demonstrations for training. On StackCube and PegInsertionSide, we scale up the position(x,y,z) 10 times and normalize the degree of griper opening for more stable learning. We set the clearance of the hole to 0.01 in PegInsertionSide so that the peg could be inserted more easily.

# F    More Experiments

## F.1    More Baselines

Generative Adversarial Imitation Learning (**GAIL**) (Ho and Ermon, 2016; Arulkumaran and Lillrank, 2024) adopts an adversarial learning framework where a discriminator is trained to differentiate between expert and agent trajectories; the discriminator's output is then used as the reward signal for the agent. Primal Wasserstein Imitation Learning (**PWIL**) (Dadashi et al., 2020; Arulkumaran and Lillrank, 2024) formulates imitation as a primal optimization problem that minimizes the Wasserstein distance between expert and agent trajectory distributions. It constructs a shaped reward function directly from this distance, encouraging the agent to produce expert-like behaviors. Soft Q Imitation Learning (**SQIL**) (Reddy et al., 2019) simplifies imitation learning by assigning fixed rewards to expert and agent transitions, effectively transforming imitation into a standard reinforcement learning problem with sparse binary rewards. Value-based Distribution Correction Estimation (**ValueDice**) (Kostrikov et al., 2019) takes a distribution-matching perspective by minimizing a divergence between expert and agent state-action occupancy measures, providing a principled connection between imitation and value-based reinforcement learning. Reinforcement Learning with Prior Data (**RLPD**) (Ball et al., 2023) is a state-of-the-art baseline improving the efficiency of online reinforcement learning by leveraging offline data. We run these baselines over 5 random experimental seeds and report the average success rate.

As the results shown in Figure 9, Cago generally outperforms all additional baseline approaches across a diverse set of manipulation tasks. In tasks such as Disassemble, StickPull, Hammer, and Pen, Cago demonstrates significantly faster convergence and higher final success rates, indicating its superior learning efficiency and robustness. Particularly in Maniskill environments, Cago is the only method that achieves meaningful learning progress, while all baselines fail to get any success, highlighting the importance of capability-aware goal sampling in challenging, sparse-reward environments. Although GAIL achieves the best performance in the ShelfPlace environment, this success is not representative of its overall effectiveness. In all other tasks, GAIL performs poorly, exhibiting highly unstable learning process and low final success rates.

## F.2    Visual goal predictor

To evaluate the generalization ability of our visual goal predictor $\mathcal{P}\phi$, we visualize its predicted goal images given initial observations at test time in Figure 10. For each task, we compare the predicted goal (middle column) to the ground-truth final observation from a demonstration trajectory (right column). Importantly, these demonstrations are collected from unseen seeds that were never used during training. These results show that $\mathcal{P}\phi$ is capable of accurately inferring the final goal state purely from a single initial image observation, even in unseen evaluation seeds. This predictive capability allows the goal-conditioned policy $\pi^G$ to finish tasks effectively for any environment seeds.

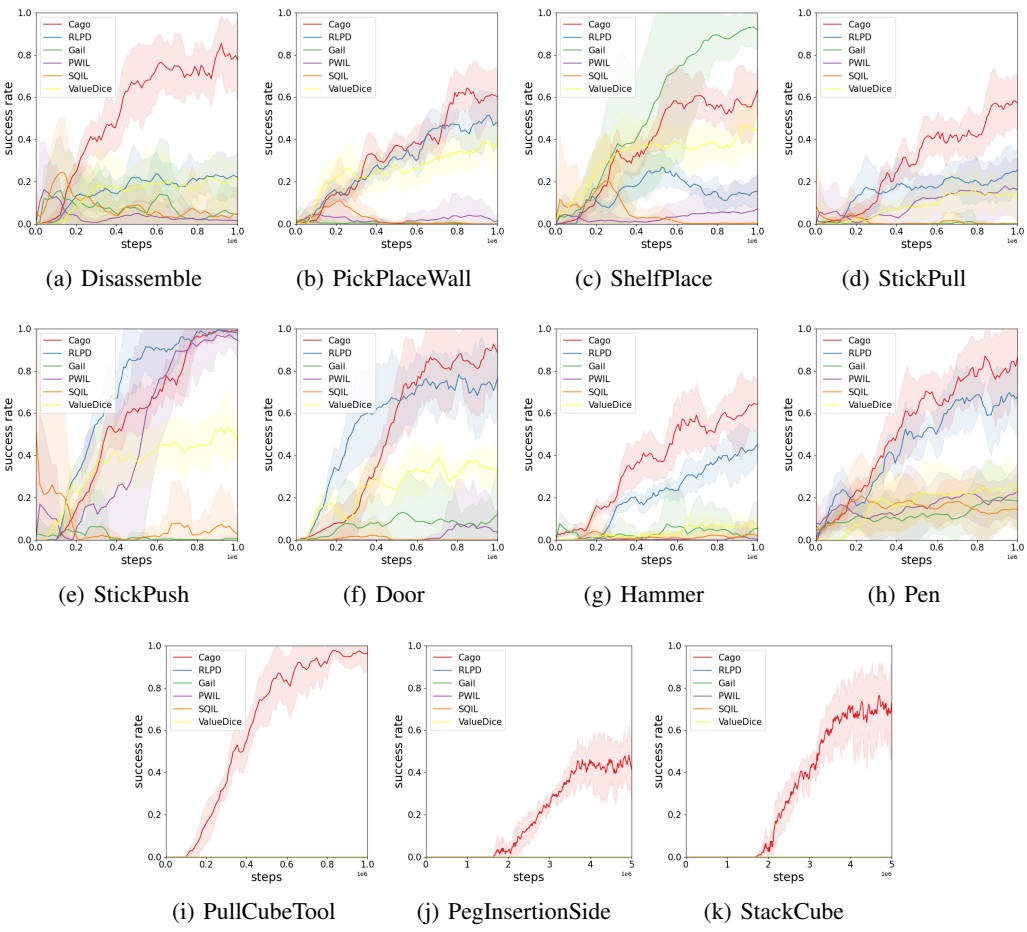

Figure 9: Experiment results comparing Cago with the additional baselines over 5 random seeds.

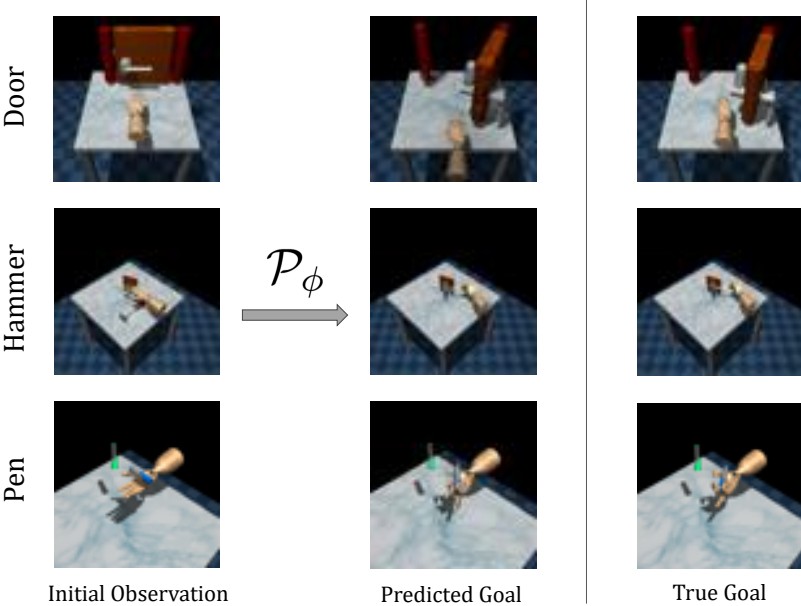

Figure 10: Visual goal prediction results from our learned goal predictor $\mathcal{P}_\phi$. Each row corresponds to a different task (Door, Hammer, Pen). From left to right: the agent's initial observation at test time, the goal image predicted by $\mathcal{P}_\phi$, and the ground-truth final observation from a demonstration trajectory. Notably, these demonstrations are drawn from unseen seeds not used during training. The predicted goals closely match the actual final states, illustrating strong generalization of $\mathcal{P}_\phi$ to novel environment seeds.

### F.3 Ablation on Number of Demonstration

This section investigates how the number of demonstration trajectories used for training influences the performance of Cago across various tasks. We present success rate curves under different numbers of demonstrations for four representative environments from Metaworld, Adroit, and ManiSkill. The results show that Cago maintains strong performance even with a limited amount of expert data (as few as 5 demonstrations), particularly in tasks such as Disassemble, StickPush, and Pen, demonstrating its robustness and stability under data-scarce conditions. Increasing the number of demonstrations in general yield higher final success rates. This effect is especially evident in more challenging tasks, such as the PegInsertionSide task in Maniskill. These findings suggest that Cago's performance can be further improved with sufficient expert demonstrations.

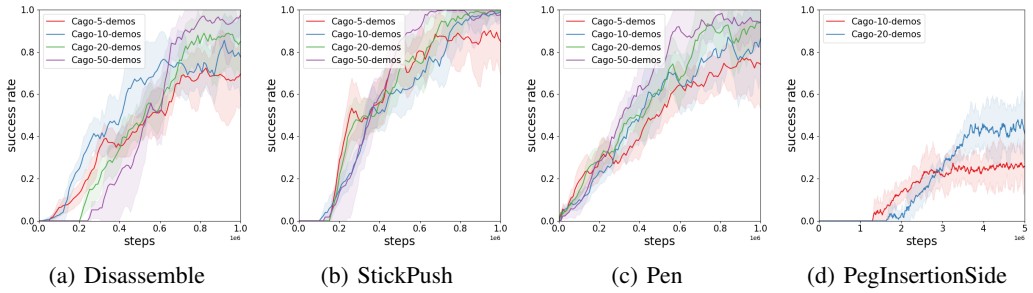

| (a) Disassemble | (b) StickPush | (c) Pen | (d) PegInsertionSide |

Figure 11: Success rates under different numbers of demonstrations. Results are averaged over 5 random seeds.

### F.4 Ablation on Demonstration Quality

A common concern in demonstration-based reinforcement learning is the reliance on high-quality trajectories that adequately cover the task space. While our theoretical analysis (Theorem 1) assumes such demonstrations in order to establish guarantees, real-world data are often noisy, incomplete, or even contain failed attempts. To evaluate the robustness of Cago under such imperfections, we conducted ablation studies on both *suboptimal* and *failed* demonstrations.

**Suboptimal Demonstrations.** We first constructed three types of perturbed demonstrations: (i) *Missing Observations*: We randomly removed 20% of the observations from each original demonstration trajectory to simulate incomplete data. (ii) *Noisy Actions*: We added Gaussian noise (scaled by 0.1) to each action output by the expert policy during expert trajectory collection. (iii) *Random Actions*: We replaced the expert's action with a randomly sampled action with a probability of 20% at each timestep during expert trajectory collection.

These settings simulate three kinds of suboptimality: missing state information, noisy expert behavior, and corrupted decision-making. Using these demonstrations, we re-evaluated Cago on three benchmark environments: AdroitPen, Stick-Push, and Disassemble. Because we do not have the expert policy for Adroit, we only conducted the "Missing Obs" experiment for Adroit-Pen. Table 4 reports the final success rates after 1M training steps, averaged over 5 random seeds (each evaluated over 100 test episodes). The results suggests that Cago is robust to the suboptimality in demonstrations. This is because: First, Cago only uses demonstration trajectories to extract sequences of observations that serve as curriculum goals. It does not attempt to imitate the demonstration directly, nor does it rely on demonstrations to infer a reward function. This significantly reduces Cago's dependency on high-quality demonstrations. Second, even when the demonstration trajectories are suboptimal—as long as they represent a successful sequence of observations—Cago can still learn well. We observe that in the Disassemble environment, Cago achieves significantly higher success rates when using the "Random Actions" type of suboptimal demonstrations compared to using the original demonstrations. Those imperfect demonstrations—with pauses or detours—cover a broader region of the task space, thereby improving the final policy's generalization. More importantly, Cago is guided by a reward function shaped using a Temporal Distance Network, which estimates how many steps are needed to reach a goal state. This enables Cago to discover more efficient, often shorter, alternative paths for goal reaching. Therefore, even when the demonstration are suboptimal, Cago may still learn the optimal paths.

Table 4: Success rates under suboptimal demonstrations (after 1M steps, averaged over 5 seeds).

| Environment | Original Demo | Missing Obs | Noisy Actions | Random Actions |
|---|---|---|---|---|
| StickPush | 0.99 | 0.99 | 0.97 | 0.96 |
| Disassemble | 0.80 | 0.77 | 0.88 | 0.96 |
| Adroit-Pen | 0.82 | 0.88 | – | – |

**Failed Demonstrations.** We next considered demonstrations that include failed trajectories. For MetaWorld, we injected uniform noise (scale 0.3) into expert actions, producing datasets with 30% and 50% failed demonstrations. We then compared Cago against several strong baselines (GAIL, PWIL, JSRL, and Modem) on Adroit-Pen, Stick-Push, and Disassemble. We assume the environments used to collect demonstrations provide a sparse (binary) reward for goal reaching, enabling us to train goal predictors exclusively on successful demonstrations. The results are shown in Figure 12, 13.

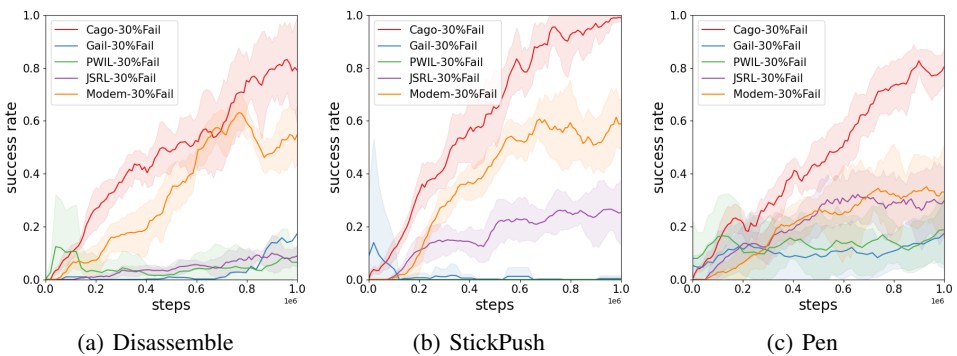

(a) Disassemble     (b) StickPush     (c) Pen

Figure 12: Success rates under 30% failed demonstrations (after 1M steps, averaged over 5 seeds).

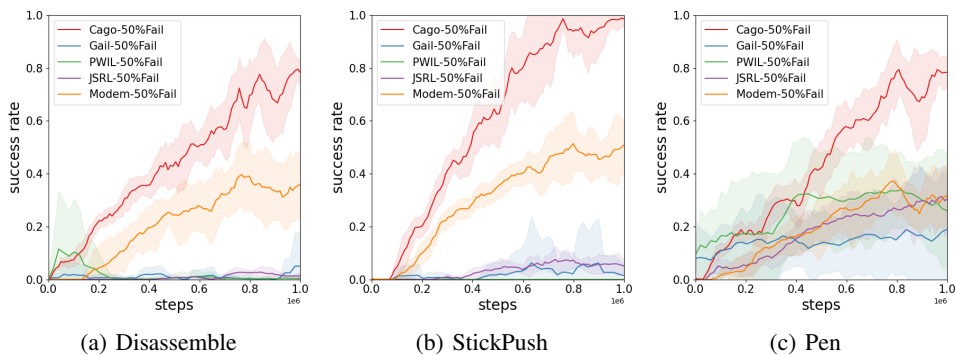

(a) Disassemble     (b) StickPush     (c) Pen

Figure 13: Success rates under 50% failed demonstrations (after 1M steps, averaged over 5 seeds).

Cago maintains strong performance even with many failed demonstrations, while the baselines degrade significantly. This robustness to suboptimal or noisy data stems from two key factors: (1) Cago prioritizes goals that are both challenging and feasible, regardless of whether they originate from successful or failed demonstrations. While failed trajectories may not directly lead to task success, they can expose the agent to diverse regions of the environment, potentially aiding exploration. This allows Cago to extract useful learning signals from a broader range of experiences. (2) Cago neither imitates demonstrations nor infers reward functions from them. Instead, it uses demonstrations solely to extract sequences of observations that serve as candidate goals. These goals are selected based on the agent's own learning progress, forming a curriculum that guides policy improvement. This design reduces sensitivity to demonstration quality.

## F.5 Ablation on Parameters

$\lambda_{visit}$ controls the threshold for how frequently a state must be visited before it is considered "mastered" by the agent, and thus eligible for sampling more difficult goals. If it is set too small, Cago may prematurely progress to harder goals before acquiring sufficient competence in easier ones. Conversely, if it is set too large, Cago may become overly conservative, spending excessive time on already-mastered states. We have conducted a sensitivity analysis by setting $\lambda_{visit} \in 50, 100, 200$, and evaluated Cago on three environments: Stick-Push, Disassemble, and AdroitPen. The final success rates after 1M training steps, averaged over 5 random seeds are shown in Table 5.

Table 5: Performance with varying $\lambda_{visit}$.

| Environment | $\lambda_{visit} = 50$ | $\lambda_{visit} = 100$ | $\lambda_{visit} = 200$ |
|---|---|---|---|
| StickPush | 0.94 | 0.96 | 0.99 |
| Disassemble | 0.79 | 0.82 | 0.80 |
| Adroit-Pen | 0.77 | 0.82 | 0.83 |

$\delta$ defines the goal sampling window size, i.e., the proportion of expert states from which new goals are sampled during training. A large $\delta$ value may lead to sampling overly distant or inappropriate goals, affecting stability and learning efficiency. In contrast, a very small $\delta$ may overly restrict exploration and hinder curriculum progression. We added a sensitivity experiment comparing the default $\delta = 10\%$ with a smaller value $\delta = 5\%$ and a larger value $\delta = 20\%$, across the same three environments. Table 6 shows that while larger values can sometimes slightly reduce performance, Cago remains capable of learning effective policies in all settings.

Table 6: Performance with varying $\delta$.

| Environment | $\delta = 5\%$ | $\delta = 10\%$ | $\delta = 20\%$ |
|---|---|---|---|
| StickPush | 1.00 | 0.99 | 0.99 |
| Disassemble | 0.81 | 0.80 | 0.75 |
| Adroit-Pen | 0.76 | 0.82 | 0.81 |

# G  Runtime

## G.1  Experiment total runtimes

Table 7: Runtimes per experiment.

| Environment | Runtime (h) | Benchmark | Steps |
|---|---|---|---|
| ShelfPlace | 72 | MetaWorld | 1e6 |
| Disassemble | 72 | MetaWorld | 1e6 |
| StickPull | 72 | MetaWorld | 1e6 |
| StickPush | 72 | MetaWorld | 1e6 |
| PickPlaceWall | 72 | MetaWorld | 1e6 |
| Adroit-Door | 80 | Adroit | 1e6 |
| Adroit-Hammer | 85 | Adroit | 1e6 |
| Adroit-Pen | 83 | Adroit | 1e6 |
| PullCubeTool | 78 | ManiSkill | 1e6 |
| PegInsertionSide | 143 | ManiSkill | 5e6 |
| StackCube | 155 | ManiSkill | 5e6 |

### G.2 Computation Time for Updating the Cago Visitation Record Dictionary

In this section, we analyze the computational cost associated with updating the visitation record dictionary. Let the length of a sampled trajectory be denoted as $L_\tau$, and let $\tau^{(i)}$ represent the demonstration trajectory associated with the same environment reset seed, having length $L_i$. The visitation record dictionary $\text{Dict}_{\text{visit}}$ is updated according to Equation 2:

$$\text{Dict}_{\text{visit}}[\tau^{(i)}][j] \mathrel{+}= 1 \quad \text{if } \text{sim}(s_t, s_j^{(i)}) \leq \epsilon, \quad \forall t \in 1, \ldots, L_\tau, \forall j \in 1, \ldots, L_i$$

This update rule implies that for each step in the sampled trajectory, a similarity check is performed against all steps in the corresponding demonstration trajectory. Thus, the time required to perform an update of $\text{Dict}_{\text{visit}}$ can be approximated by:

$$\text{Time(Update)} \approx \text{Total Steps} \times L_i \times \text{Time(Similarity Calculation)}$$

The computational cost is therefore influenced by three main factors: the total number of interaction steps, the length of each demonstration trajectory, and the cost of computing the similarity metric. Importantly, the similarity function $\text{sim}(\cdot, \cdot)$ differs by environment, which directly affects computation time. For MetaWorld environments, we utilize L2-distance in the state vector space (i.e., low-dimensional numerical vectors). This calculation is computationally efficient, typically requiring only simple element-wise operations over vector entries. In contrast, for the Adroit and Maniskill environments, the similarity is computed based on MSE in the image space. This involves pixel-wise comparison over image observations, which increases the computational load due to the large input dimensionality (e.g., $100 \times 100 \times 3$). As a result, while the update rule remains structurally the same, the actual runtime overhead for image-based similarity can be substantially higher than that for state-based similarity. The table below summarizes the total runtime, update time, and similarity function used for each environment:

Table 8: Computation time and similarity function for updating the visitation dictionary $\text{Dict}_{\text{visit}}$.

| Environment | Steps | Runtime (h) | Update Time (h) | Similarity |
|---|---|---|---|---|
| Disassemble | 1e6 | 72 | 0.05 | State L2 |
| PickPlaceWall | 1e6 | 72 | 0.05 | State L2 |
| ShelfPlace | 1e6 | 72 | 0.05 | State L2 |
| StickPull | 1e6 | 72 | 0.05 | State L2 |
| StickPush | 1e6 | 72 | 0.05 | State L2 |
| Adroit-Door | 1e6 | 80 | 5.3 | Image MSE |
| Adroit-Hammer | 1e6 | 85 | 5.3 | Image MSE |
| Adroit-Pen | 1e6 | 83 | 5.3 | Image MSE |
| PullCubeTool | 1e6 | 78 | 2.8 | Image MSE |
| PegInsertionSide | 5e6 | 143 | 13.8 | Image MSE |
| StackCube | 5e6 | 155 | 14.2 | Image MSE |

## H  Hyperparameters

We adopt the default hyperparameters from the LEXA backbone model-based RL (MBRL) agent—such as the learning rate, optimizer, and network architecture—and maintain them consistently across all environments. The primary hyperparameter tuning for Cago focuses on the following aspects: (1) the episode length $L_\tau$; (2) the proportion of $L_\tau$ allocated to the goal-directed phase $T_{go}$; (3) the number of demonstrations $N_{demo}$ used for both dictionary construction and environment resetting; (4) the visit frequency threshold $\lambda_{visit}$ used in Algorithm 1 for filtering goal candidates; and (5) the similarity calculate metrics in Equation 2; (6) the similarity threshold $\epsilon$ in Equation 2.

Table 9: Hyperparameters of Cago.

| Environment | $L_\tau$ | $T_{go}\ rate$ | $N_{demo}$ | $\lambda_{visit}$ | $\delta$ | $sim(\cdot,\cdot)$ | $\epsilon$ |
|---|---|---|---|---|---|---|---|
| Disassemble | 300 | 0.6 | 10 | 200 | 10% | State L2 | 0.1 |
| PickPlaceWall | 150 | 0.6 | 10 | 200 | 10% | State L2 | 0.08 |
| ShelfPlace | 150 | 0.6 | 10 | 50 | 10% | State L2 | 0.05 |
| StickPull | 200 | 0.6 | 10 | 200 | 10% | State L2 | 0.08 |
| StickPush | 150 | 0.7 | 10 | 200 | 10% | State L2 | 0.05 |
| Adroit-Door | 200 | 0.7 | 10 | 100 | 10% | Image MSE | 200.0 |
| Adroit-Hammer | 100 | 0.6 | 10 | 200 | 10% | Image MSE | 50.0 |
| Adroit-Pen | 100 | 0.7 | 10 | 200 | 10% | Image MSE | 200.0 |
| PullCubeTool | 100 | 0.7 | 10 | 100 | 10% | Image MSE | 100.0 |
| PegInsertionSide | 100 | 0.7 | 20 | 100 | 10% | Image MSE | 100.0 |
| StackCube | 100 | 0.7 | 20 | 100 | 10% | Image MSE | 50.0 |

# NeurIPS Paper Checklist

The checklist is designed to encourage best practices for responsible machine learning research, addressing issues of reproducibility, transparency, research ethics, and societal impact. Do not remove the checklist: **The papers not including the checklist will be desk rejected.** The checklist should follow the references and follow the (optional) supplemental material. The checklist does NOT count towards the page limit.

Please read the checklist guidelines carefully for information on how to answer these questions. For each question in the checklist:

- You should answer [Yes] , [No] , or [NA] .
- [NA] means either that the question is Not Applicable for that particular paper or the relevant information is Not Available.
- Please provide a short (1–2 sentence) justification right after your answer (even for NA).

**The checklist answers are an integral part of your paper submission.** They are visible to the reviewers, area chairs, senior area chairs, and ethics reviewers. You will be asked to also include it (after eventual revisions) with the final version of your paper, and its final version will be published with the paper.

The reviewers of your paper will be asked to use the checklist as one of the factors in their evaluation. While "[Yes] " is generally preferable to "[No] ", it is perfectly acceptable to answer "[No] " provided a proper justification is given (e.g., "error bars are not reported because it would be too computationally expensive" or "we were unable to find the license for the dataset we used"). In general, answering "[No] " or "[NA] " is not grounds for rejection. While the questions are phrased in a binary way, we acknowledge that the true answer is often more nuanced, so please just use your best judgment and write a justification to elaborate. All supporting evidence can appear either in the main paper or the supplemental material, provided in appendix. If you answer [Yes] to a question, in the justification please point to the section(s) where related material for the question can be found.

IMPORTANT, please:

- **Delete this instruction block, but keep the section heading "NeurIPS Paper Checklist",**
- **Keep the checklist subsection headings, questions/answers and guidelines below.**
- **Do not modify the questions and only use the provided macros for your answers**.

