# OpenReview forum: "Learning from Demonstrations via Capability-Aware Goal Sampling"
_NeurIPS.cc/2025/Conference — NeurIPS 2025 poster_

### Official Review · Reviewer_NtvZ · 2025-06-21

**Clarity:** 4
**Significance:** 2
**Originality:** 3
**Rating:** 5
**Confidence:** 4

**Summary:**

The paper presents Capability-Aware Goal Sampling (Cago), a new method for imitation learning in tasks that require reaching goals over many steps. Cago works by first picking goals that are just beyond what the agent can currently do, helping it learn in manageable steps. In the next phase, an exploration policy trained from demonstrations helps the agent explore the environment. The approach also uses model-based reinforcement learning, where a world model and the goal-conditioned policy are trained together. Experiments show that Cago leads to much better sample efficiency and final performance than current leading methods.

**Questions:**

In addition to the questions raised in the weaknesses section, addressing the following points would help alleviate my concerns:
+ Why not initialize the goal-conditioned policy with behavioral cloning (BC) pre-training, for example by sampling future states as goals?
+ How does the method’s performance scale with the number of demonstrations? Improving the BC policy and the $L_\tau$ component in Algorithm 2 could potentially make the online phase much more efficient. Given the high training cost of your method, it would be helpful to justify the choice of using only 10–20 demonstrations.

**Ethical Concerns:**

["NO or VERY MINOR ethics concerns only"]

**Final Justification:**

After carefully considering the authors' rebuttal and the discussion among reviewers, I have updated my assessment as follows:

**Resolved Issues:**
- The authors provided comprehensive ablation studies and analyses clarifying the role and robustness of each module in their framework, which addressed my concerns about the method's complexity and the importance of individual components.
- They gave detailed responses on the method’s applicability to broader, real-world scenarios, including diverse initial states and stochastic dynamics.
- The rationale for not using behavioral cloning (BC) pre-training was explained clearly, and the approach was justified in the context of sparse demonstrations.
- Additional experiments and ablations on the number of demonstrations demonstrated that the method is effective even with limited expert data, which is important for practical deployment.

**Remaining Issues:**
- While the authors addressed most technical questions, some open questions remain regarding the method's scalability to highly abstract or non-robotic tasks and the potential overhead in real-world deployment.
- The paper could still benefit from deeper qualitative analysis and a more explicit discussion of limitations in the main text.

**Overall Assessment:**
The rebuttal has largely addressed my main concerns, especially regarding robustness, practical applicability, and experimental validation. The remaining issues are relatively minor and do not outweigh the strengths demonstrated. Therefore, I have raised my score to reflect the improved confidence in the paper's contribution.

**Limitations:**

The discussion on how the method could be applied to broader scenarios—such as real-world settings with diverse initial states and uncertain dynamics—could be expanded.

**Quality:**

3

**Strengths And Weaknesses:**

**Strengths:**
+ The paper clearly defines the problem: existing imitation learning methods struggle with long-horizon tasks due to compounding errors and a lack of awareness of the agent’s own capabilities. The descriptions of the proposed method and the experimental setup are also clear and well-organized.
+ The experiments demonstrate significant improvements over baseline methods.

**Weaknesses:**
+ The approach involves several components—goal sampling, goal-conditioned policy, behavioral cloning (BC) policy, world model, and goal predictor. It would be helpful to understand how robust the overall method is to the performance of each individual component.
+ The discussion on how the method could be applied to broader scenarios—such as real-world settings with diverse initial states and uncertain dynamics—could be expanded. This would help clarify the method’s generalizability and practical impact.

---

> ### Author Rebuttal · Authors · 2025-07-31
>
> We appreciate your insightful feedback and constructive comments!
>
> **W1. The approach involves several components. It would be helpful to understand how robust the overall method is to each individual component.**
>
> Cago consists of five main modules: (1) a capability-aware goal sampler, (2) a BC explorer, (3) a goal-conditioned policy, (4) a world model, and (5) a goal predictor. At a high level, Cago trains the world model and goal-conditioned policy in an end-to-end manner. The world model is updated using real trajectories collected by the goal-conditioned policy, whose goal is set by the capability-aware goal sampler (Algorithm 2, Line 18). Upon goal reaching, a pretrained BC explorer takes over for the remaining time steps in a rollout. Simultaneously, the goal-conditioned policy is trained using imagined rollouts generated by the world model (Line 19). These two components are tightly coupled and co-adapt during training, making their joint performance critical to overall success. The goal predictor is trained separately on demonstrations to infer the final goal state from an initial observation. It is used only during evaluation and enables the goal-conditioned policy to generalize to test environments where the goal state is not explicitly provided.
>
> We assess the importance of each component as follows.
>
> (1) To ablate the capability-aware goal sampler, we designed two additional goal sampling strategies:
>
> * Step-based curriculum: This baseline samples goals from demonstrations in proportion to training steps (i.e., current training step / predefined total training steps). However, it does not assess the agent's actual capabilities, and may therefore sample goals that are either too easy or too difficult for the agent at its current learning stage.
> * Final-goal sampling (Line 329): This strategy always selects the final observation from a demonstration in the goal phase of our Go-Explore sampling paradigm, ignoring the agent's current goal-reaching capability. The BC Explorer takes control from the goal-conditioned policy halfway through each rollout. The table below reports the final success rates averaged over 5 random seeds (each evaluated over 100 test episodes):
>
> | Environment     | Cago (Ours) | Step-based Curriculum | Final-goal Sampling |
> |----------------|-------------|------------------------|---------------------|
> | Adroit-Pen     |    **0.82**         |           0.48             |          0.52           |
> | Stick-Push     |      **0.99**       |            0.92            |          0.68           |
> | Disassemble    |      **0.80**       |           0.04             |           0.02          |
>
> Cago's goal sampling strategy significantly outperforms the alternatives, highlighting its critical role in the overall effectiveness of our approach.
>
> (2) As for BC-Explorer, we provide detailed ablation studies in Section 4 (Line 328). Specifically, Cago-NoExplorer removes the BC Explorer. It does not further explore after reaching a sampled goal. We also evaluate Cago-RandomExplorer, which replaces the BC Explorer with a uniformly random policy during the exploration phase of our Go-Explore-style rollout strategy. The results are in Fig. 4. The table below reports the final success rates averaged over 5 random seeds:
>
> | Environment     | Cago-BC-Explorer (Ours) | Cago-NoExplorer | Cago-RandomExplorer |
> |----------------|-------------|------------------------|---------------------|
> | Adroit-Pen     |    **0.82**         |           0.60             |          0.69           |
> | Stick-Push     |      **0.99**       |            0.72            |          0.79           |
> | Disassemble    |      **0.80**       |           0.61             |           0.72          |
>
> The results highlight the importance of BC Explorer for generating relevant exploratory rollouts to accelerate learning.
>
> (3) For the goal-conditioned policy, we observed that its critic loss decreases over time in correlation with the success rates reported in Fig. 3, underscoring its crucial role in overall performance.
>
> (4) In early development, we attempted a model-free variant of Cago by training a goal-conditioned policy directly from environment rollouts starting at demonstration initial states using actor-critic methods. However, due to the limited number of demonstrations, this approach failed to generalize. We therefore adopted the model-based strategy that leverages a world model to generate diverse imagined rollouts (Line 173), enabling more effective and sample-efficient policy learning.
>
> (5) Empirically, we find that the Goal Predictor reliably infers final goals from held-out demonstrations, enabling effective zero-shot execution in unseen test tasks.
>
> **W2. The discussion on how the method could be applied to broader scenarios—such as real-world settings with diverse initial states and uncertain dynamics—could be expanded.**
>
> Cago can be extended to support diverse initial states and uncertain dynamics.
> * For diverse initial states in real-world settings, Cago can gradually prioritize those where the agent has the most learning potential—starting from demonstration initial states and expanding to nearby ones where the agent can still benefit from the demonstration trajectories for dynamic capability tracking and progressive goal selection. As training progresses, we incorporate successful rollouts from these nearby initial states into the demonstration set, broadening its coverage. This enables a natural curriculum that eventually transitions to diverse, real-world initial states.
> * For uncertain, stochastic dynamics, Cago remains robust by not requiring exact reproduction of actions in demonstration trajectories. Instead, it adapts goals based on observed agent successes, making the curriculum self-correcting in the face of noise or variability. If the agent consistently fails to reach a specific demonstration state, Cago's capability metric naturally shifts goal sampling toward more achievable targets. Additionally, goal selection can be extended with probabilistic estimates—such as goal-reaching likelihood or future state predication uncertainty from an ensemble of world models—which we view as a promising direction for high-stochasticity domains.
>
> **Q1. Why not initialize the goal-conditioned policy with behavioral cloning (BC) pre-training?**
>
> We considered the possibility of initializing $\pi^G$ with behavioral cloning (BC) on demonstration data, and conducted preliminary experiments with BC pretraining in Cago’s early development stage. However, our findings led us to deliberately decide against using BC initialization in our final framework, for the following reasons:
>
>  * (1) Risk of overfitting under sparse demonstrations: Cago is specifically designed for the low-data regime, where only a small number of demonstrations (typically 5 or 10) are available. In such cases, BC pretraining tends to overfit to the limited demonstration data, resulting in poor generalization when the agent starts to explore beyond the demonstrated trajectories.
>
> * (2) Misalignment with the training framework of Cago: Cago’s goal-conditioned policy $\pi^G$ is trained jointly with a Temporal Distance Network (TDN), which estimates how many steps are needed to reach a goal. The TDN serves as a dense feedback signal and is itself trained on imagined trajectories from the world model. Crucially, all three components—the policy, world model, and TDN—are updated jointly in an end-to-end manner. In the early phase of training, the TDN’s estimates are often inaccurate. Therefore, even a well-behaved BC-initialized policy would receive noisy or misleading training signals, causing it to quickly deviate from the pretraining distribution. This undermines the value of BC initialization and can sometimes even slow down learning. These observations informed our design decision to exclude BC pretraining.
>
> **Q2. How does the method’s performance scale with the number of demonstrations? ... It would be helpful to justify the choice of using only 10–20 demonstrations.**
>
> Our method is explicitly designed for the low-demonstration regime, which is typical in real-world robotic settings where expert trajectories are expensive, time-consuming, or risky to collect. We intentionally evaluate Cago with only 10–20 demonstrations to highlight its practical relevance, emphasizing sample efficiency and generalization from limited supervision, enabled by the use of a world model to generate imagined rollouts that enrich training. A more accurate model around the demonstration distribution can benefit policy generalization. We conducted ablation study in the Supplemental Material (Figure 1)  using 5, 10, and 20 demonstrations. We further conducted an experiment using 50 demonstrations to evaluate how Cago performs under a large set of expert data. The table below reports the final success rates averaged over 5 random seeds (each evaluated over 100 test episodes):
>
> | Environment | Cago-5-demons | Cago-10-demons | Cago-20-demons | Cago-50-demons |
> |----------------|-------------|------------------------|---------------------|---------------------|
> | Adroit-Pen     |    0.71         |           0.82             |          0.92           |          **0.93**           |
> | Stick-Push     |      0.81       |            0.99            |          1.00           |          **1.00**           |
> | Disassemble    |      0.69       |           0.80             |           0.81          |          **0.98**           |
>
> We found that with 50 demonstrations, Cago performs exceptionally well, exhibiting significantly better generalization and achieving higher final success rates compared to training with fewer demonstrations. However, we also found that larger demonstration sets may delay learning onset due to longer warm-up times to fit world models. Studying this trade-off is an interesting direction.

---

> > ### Comment · Reviewer_NtvZ · 2025-08-01
> >
> > The authors' rebuttal has largely addressed my concerns. I will raise my score accordingly.

---

> > > ### Author Response · Authors · 2025-08-06
> > > **Thank you for your feedback!**
> > >
> > > We truly appreciate your acknowledgment of our rebuttal and are grateful for your continued support. We will incorporate the suggested ablation study, the discussion on broader applicability, the trade-off of initializing the goal-conditioned policy with behavioral cloning, and the analysis of how Cago scales with the number of demonstrations into the final revision. These suggestions have significantly improved the quality and clarity of our paper!

---

### Official Review · Reviewer_1nz5 · 2025-06-23

**Clarity:** 3
**Significance:** 3
**Originality:** 2
**Rating:** 4
**Confidence:** 4

**Summary:**

This paper proposes Cago (Capability-Aware Goal Sampling), a novel method to enhance the efficiency of Imitation Learning (IL) in long-horizon, sparse-reward environments. Cago dynamically tracks the agent's current capabilities and, based on this, intelligently samples intermediate goals from expert demonstrations that are just beyond the agent's current reach, thereby adaptively adjusting the learning curriculum. This approach aims to mitigate the limitations of fixed curricula or direct imitation of demonstrations, such as error accumulation and inefficient exploration of the task space. The proposed framework integrates a Go-Explore paradigm, which leverages goal-conditioned policies and behavior cloning explorers, along with a predictive world model, enabling efficient data collection and policy optimization. The paper presents experimental results demonstrating Cago's superior performance compared to existing IL baselines across various robotic manipulation tasks.

**Questions:**

1. Real-world demonstration data inevitably contains noise or can be incomplete. This paper assumes high-quality demonstrations. Have you conducted experimental analyses to see how Cago's performance changes when the demonstration data is intentionally noised (e.g., deleting some observations, using demonstrations with unclear target states), when the expertise of the demonstrator is reduced, or when the number of demonstrations is severely limited? Without such analysis, it is difficult to be convinced of Cago's utility in real-world environments.
2. The `Dict_visit`based capability tracking and goal sampling mechanism might work well in clear state spaces like robot arm positions. However, in situations where 'goal states' are difficult to define clearly within demonstration trajectories, such as tasks involving linguistic instructions or needing to set goals within high-dimensional, abstract visual feature spaces, how can Cago sample goals and track capabilities?
3. The values of key hyperparameters like `lambda_visit` and `delta` are critically important for Cago's adaptive curriculum. A detailed sensitivity analysis on how changes in these values affect learning efficiency and final performance should be included in the paper. Currently, there is a lack of clear explanation on how these parameters were determined, and no guidelines are provided on how to tune them for new tasks.

**Ethical Concerns:**

["NO or VERY MINOR ethics concerns only"]

**Final Justification:**

The additional experiments and clarifications have improved the submission in several aspects, including new results under missing and noisy demonstrations, analyses of hyperparameters, and inclusion of PEG as a GCRL baseline. However, some core limitations remain only partially addressed—most notably, the reliance on successful demonstrations and the limited scalability to high-dimensional or semantic goal spaces.

**Limitations:**

The authors do not adequately address the limitations of their work within the main body of the paper. While the "Checklist" in the appendix indicates that "Limitations" are discussed, a dedicated section or clear discussion within the core text is absent. This makes it difficult for readers to quickly identify and understand the boundaries and potential shortcomings of the proposed method.

**Paper Formatting Concerns:**

The paper generally adheres to NeurIPS 2025 formatting guidelines. The structure is clear, and figures and tables are appropriately placed. No major formatting concerns were identified.

**Quality:**

2

**Strengths And Weaknesses:**

### Strengths

- The core idea of "Capability-Aware Goal Sampling" is itself interesting and represents a novel attempt to address the limitations of imitation learning. The adaptive curriculum approach, which dynamically sets goals based on the agent's capabilities, is theoretically appealing.
- The construction of a framework that integrates multiple components, such as observation visit tracking, the Go-Explore paradigm, and a world model, demonstrates the complex design of the methodology.
- The attempt to evaluate Cago's performance across various robotic manipulation environments showcases an effort to demonstrate the methodology's applicability.

### Weaknesses

- This methodology critically relies on "high-quality demonstration trajectories that adequately cover the task space." While mentioned as a limitation in the paper, this core assumption is neither sufficiently justified nor mitigated, considering the significant difficulty in reliably obtaining high-quality demonstrations in the real world. The methodology severely lacks robustness to noisy, incomplete, or suboptimal expert demonstrations, with no experimental analysis or proposed solutions provided. This fundamentally limits the practical applicability and generality of the proposed method.
- The `Dict_visit`based capability tracking and visit-count-based goal sampling appear tailored for environments where goals can be clearly defined as states directly from demonstrations. However, there is a clear lack of discussion or evidence regarding its scalability to highly-dimensional or abstract continuous goal spaces, or situations where goals are difficult to define directly from demonstration trajectories (e.g., abstract goals like "achieve emotional state"). Without in-depth analysis in this area, the general utility of the methodology remains low.
- Key hyperparameters in Cago, such as `lambda_visit` (visitation frequency threshold) and `delta` (window size for goal sampling), are crucial in determining the behavior of the adaptive curriculum. However, there is an absence of an in-depth sensitivity analysis on how these parameters affect performance. The paper lacks a clear explanation of how misconfigurations of these parameters might disrupt the learning process (e.g., repeatedly selecting only overly easy goals, or consistently pursuing unattainable goals), leading to significant uncertainty in applying the method to new tasks.
- Although Cago is compared to demonstration-based baselines, it overlooks comparisons with broader goal-conditioned reinforcement learning methods beyond demonstrations. This weakens the clarity of Cago's distinct contributions and its unique advantages over existing approaches.
- Although quantitative results are presented, a deeper qualitative analysis of *how* Cago's learned policy behaves during the exploration phase, and how Cago succeeds in specific scenarios where baselines fail, is missing. This hinders a comprehensive understanding of the methodology's internal workings and the fundamental reasons for its success.

---

> ### Author Rebuttal · Authors · 2025-07-31
>
> We appreciate your insightful feedback and constructive comments!
>
> **1. This methodology critically relies on "high-quality demonstration trajectories that adequately cover the task space." The methodology lacks experimental analysis or solutions to noisy, incomplete, or suboptimal expert data.**
>
> We clarify that Cago only assumes access to "high-quality demonstration trajectories that adequately cover the task space" (Line 246) in order to derive the theoretical guarantee stated in Theorem 1. In practice, the demonstrations used in the Adroit environments (door, hammer, and pen) are in fact noisy and suboptimal, and yet Cago remains effective. This highlights the robustness of our method to imperfect demonstrations.
>
> To further address this concern, we have conducted additional experiments specifically designed to evaluate how Cago performs when provided with noisy, incomplete demonstrations. We collected three different types of suboptimal demonstrations:
>
> * (1) Missing Observations: We randomly removed 20% of the observations from each original demonstration trajectory to simulate incomplete data.
>
> * (2) Noisy Expert Actions: We added Gaussian noise (scaled by 0.1) to each action output by the expert policy during expert trajectory collection.
>
> * (3) Random Actions: We replaced the expert’s action with a randomly sampled action with a probability of 20% at each timestep during expert trajectory collection.
>
> These settings simulate three kinds of suboptimality: missing state information, noisy expert behavior, and corrupted decision-making. Using these demonstrations, we re-evaluated Cago on three benchmark environments: AdroitPen, Stick-Push, and Disassemble. Because we do not have the expert policy for Adroit, we only conducted the "Missing Obs" experiment for Adroit-Pen. The table below reports the final success rates after 1M training steps, averaged over 5 random seeds (each evaluated over 100 test episodes):
>
> | Environment     | Original Demo | Missing Obs | Noisy Actions | Random Actions |
> |----------------|-------------------|--------------------|------------------------|----------------------|
> | Adroit-Pen     | 0.82                |  **0.88**               |  -                    |   -                 |
> | Stick-Push     | **0.99**                 | **0.99**                  | 0.97                      | 0.96                    |
> | Disassemble    | 0.80                 | 0.77                  | 0.88                      | **0.96**                    |
>
>  The results suggests that Cago is robust to the suboptimality in demonstrations. This is because:
>
> * First, Cago only uses demonstration trajectories to extract sequences of observations that serve as curriculum goals. It does not attempt to imitate the demonstration directly, nor does it rely on demonstrations to infer a reward function. This significantly reduces Cago's dependency on high-quality demonstrations.
>
> * Second, even when the demonstration trajectories are suboptimal—as long as they represent a successful sequence of observations—Cago can still learn well. We observe that in the Disassemble environment, Cago achieves significantly higher success rates when using the "Random Actions" type of suboptimal demonstrations compared to using the original demonstrations. Those imperfect demonstrations—with pauses or detours—cover a broader region of the task space, thereby improving the final policy's generalization. More importantly, Cago is guided by a reward function shaped using a Temporal Distance Network (see Main Paper Appendix B.2), which estimates how many steps are needed to reach a goal state. This enables Cago to discover more efficient, often shorter, alternative paths for goal reaching. Therefore, even when the demonstration are suboptimal, Cago may still learn the optimal paths. We will revise the paper to highlight Cago's robustness to suboptimal demonstrations by including these results.
>
> **2. Here is a clear lack of discussion or evidence regarding its scalability to situations where goals are difficult to define directly from demonstration trajectories.**
>
> Our work specifically targets robotic manipulation tasks, where it is both common and effective to define the goal space directly in terms of the observation space. This approach has been widely adopted in prior works on goal-conditioned reinforcement learning [1,2,3]. In Appendix F.2, we demonstrate that Cago performs comparably in high-dimensional visual environments, even when using a simple similarity metric such as mean squared error (MSE) between image-based observations. Since Cago guides agents toward low-level demonstration states in a progressive manner, it does not rely on or infer high-level abstract goals—doing so is outside the intended scope of our method.
>
> [1] Planning goals for exploration, ICLR 2023
>
> [2] Learning World Models for Unconstrained Goal Navigation, NeurIPS 2024
>
> [3] Discovering and Achieving Goals via World Models, NeurIPS 2021
>
> **3. More sensitivity analysis on how lambda_visit and delta affect performance.**
>
> Regarding $\lambda_{visit}$: This parameter controls the threshold for how frequently a state must be visited before it is considered “mastered” by the agent, and thus eligible for sampling more difficult goals. If it is set too small, Cago may prematurely progress to harder goals before acquiring sufficient competence in easier ones. Conversely, if it is set too large, Cago may become overly conservative, spending excessive time on already-mastered states. We have conducted a sensitivity analysis by setting $\lambda_{visit} \in {50, 100, 200}$, and evaluated Cago on three environments: Stick-Push, Disassemble, and AdroitPen. The final success rates after 1M training steps, averaged over 5 random seeds are shown below.
>
> | Environment     | $\lambda_{visit}$ = 50 | $\lambda_{visit}$ = 100 | $\lambda_{visit}$ = 200 |
> |----------------|--------------|---------------|---------------|
> | Stick-Push     | 0.94         | 0.96          | **0.99**          |
> | Disassemble    | 0.79      | **0.82**          | 0.80          |
> | Adroit-Pen     | 0.77         | 0.82          | **0.83**          |
>
> Results show that while very small $\lambda_{visit}$ may slightly influence learning, Cago consistently reaches similar final performance, indicating robustness to this parameter across a broad range.
>
> Regarding $\delta$: This parameter defines the goal sampling window size, i.e., the proportion of expert states from which new goals are sampled during training. A large $\delta$ value may lead to sampling overly distant or inappropriate goals, affecting stability and learning efficiency. In contrast, a very small $\delta$ may overly restrict exploration and hinder curriculum progression. We added a sensitivity experiment comparing the default $\delta$ = 10% with a smaller value $\delta$ = 5% and a larger value $\delta$ = 20%, across the same three environments. The table below shows that while larger values can sometimes slightly reduce performance, Cago remains capable of learning effective policies in all settings. We will include these results to the paper to provide a more thorough discussion of both parameters.
>
> | Environment     | $\delta$ = 5% | $\delta$ = 10% | $\delta$ = 20% |
> |----------------|---------|---------|---------|
> | Stick-Push     | **1.00**    | 0.99    | 0.99    |
> | Disassemble    | **0.81**    | 0.80    | 0.75    |
> | Adroit-Pen     | 0.76    | **0.82**    | 0.81    |
>
> **4. Although Cago is compared to demonstration-based baselines, it overlooks comparisons with broader goal-conditioned reinforcement learning methods beyond demonstrations.**
>
> Cago introduces a novel paradigm for leveraging demonstrations, with goal-conditioned RL (GCRL) serving only as the underlying framework. Our focus is on demonstration utilization, so we adapt a broad set of strong baselines that differ in how they use demonstrations. To further address the reviewer’s concern, we include PEG [1], a recent competitive model-based GCRL method, alongside the existing GCRL-Dreamer baseline. As shown in the table below, Cago consistently outperforms these GCRL baselines.
>
> Table4: Results of Cago and GCRL baselines.
> | Environment     | Cago(Ours)| Dreamer | PEG |
> |----------------|--------------|---------------|---------------|
> | Stick-Push     |    **0.99**     |     0.68     |    0.76      |
> | Disassemble    |    **0.80**     |     0.02     |     0.04      |
> | Adroit-Pen     |    **0.82**     |      0.52    |      0.54    |
>
> [1] Hu. et al., Planning goals for exploration, ICLR 2023
>
> **5. How Cago's learned policy behaves during the exploration phase, and how Cago succeeds in specific scenarios where baselines fail.**
>
> We visualized the exploration phase of Cago for the StickPush environment in Figure 4(d). Each red dot represents the normalized position of a sampled goal within a demonstration trajectory, with 0 indicating the start and 1.0 indicating the final demonstration state. Early in training, the agent predominantly samples goals at lower normalized positions, focusing on subgoals near the beginning of the trajectory that are within its current capabilities. As training advances, goal sampling gradually shifts toward higher normalized positions, indicating the agent’s increasing ability to pursue more challenging goals closer to task completion. By continuously targeting goals just at the boundary of the agent’s current capability, Cago facilitates efficient learning in this long-horizon task.  In challenging tasks like Peg-insertion, Cago's curriculum-like progression—guided by internal capability estimation—significantly improves learning efficiency. In contrast, we found the baseline policies, while attempting to imitate the demonstrations, move the manipulator to the final pose even the it does not yet grasp the peg. Similar exploration trends are observed across other benchmarks, and we will include these figures in the revised paper.

---

> ### Comment · Reviewer_1nz5 · 2025-08-02
>
> I thank the authors for their response and the additional experiments. The additional experiments and clarifications address several of the initial concerns. However, several fundamental concerns remain unaddressed or only partially mitigated.
>
> **1. Unrealistic Demonstration Assumptions**
>
> While the additional experiments with noisy demonstrations are appreciated, all variants still assume that the demonstrations lead to successful task completion. In real-world settings, demonstrations are often incomplete, ambiguous, or include failed attempts—conditions under which Cago, as currently designed, is not equipped to operate. This limits its robustness in practical deployment scenarios.
>
> **2. Scalability to Abstract or Semantic Goal Spaces**
>
> Cago critically depends on defining goals as observable states directly sampled from demonstrations, and similarity is computed via MSE in pixel or state space. This approach is:
>
> - Fragile in high-dimensional visual settings, and
> - Inapplicable to tasks requiring language conditioning or abstract goal inference.
>
> Therefore, I am inclined to slightly raise my score to reflect the improvements, while still noting that further refinement is needed to reach broader applicability.

---

> > ### Author Response · Authors · 2025-08-06
> > **Thank you for your feedback!**
> >
> > Thank you for your constructive feedback!
> >
> > **Addressing Unrealistic Demonstration Assumptions**
> >
> > We agree that, in real-world settings, demonstrations can be ambiguous and include failed attempts. To address this concern and demonstrate the applicability of Cago in such scenarios, we conducted new experiments using noisy demonstrations that include failed trajectories. For MetaWorld, we introduced significant noise into the expert actions by adding uniform noise scaled by a factor of 0.3. This resulted in a substantial number of trajectories that fail to achieve the goal conditions. Using these, we created two new demonstration datasets containing:
> >
> > * **30% failed demonstrations**
> > * **50% failed demonstrations**
> >
> > We then evaluated Cago and several strong baselines (as reported in our paper) on three representative environments: Adroit-Pen, Stick-Push, and Disassemble. We assume the environments used to collect demonstrations provide a sparse (binary) reward for goal reaching, enabling us to train goal predictors exclusively on successful demonstrations. We compared the performance under such noisy demonstration settings against the original results reported in the paper, in order to assess robustness. The final success rates after 1M training steps, averaged over 5 random seeds are shown below. *The numbers in parentheses refer to the original results reported in the paper.*
> >
> > * 30% failed demonstrations:
> >
> > | Environment     | Cago(Ours) | Gail | PWIL | JSRL | Modem|
> > |----------------|-------------|------------------------|---------------------|---------------------|---------------------|
> > | Adroit-Pen| 0.78(0.82) ↓0.04| 0.18(0.17) ↑0.01| 0.26(0.22) ↓0.04| 0.28(0.45) ↓0.17| 0.31(0.32) ↓0.01|
> > | Stick-Push| 0.99(0.99) -  | 0.01(0.01) - | 0.74(0.92) ↓0.18| 0.24(0.37) ↓0.13| 0.60(0.75) ↓0.15|
> > | Disassemble| 0.81(0.80) ↑0.01| 0.06(0.06) - | 0.09(0.03) ↑0.06| 0.10(0.19) ↓0.09| 0.54(0.59) ↓0.05|
> >
> > * 50% failed demonstrations:
> >
> > | Environment     | Cago(Ours) | Gail | PWIL | JSRL | Modem|
> > |----------------|-------------|------------------------|---------------------|---------------------|---------------------|
> > | Adroit-Pen| 0.76(0.82) ↓0.06| 0.15(0.17) ↓0.02| 0.13(0.22) ↓0.09| 0.32(0.45) ↓0.13| 0.29(0.32) ↓0.03|
> > | Stick-Push| 0.98(0.99) ↓0.01| 0.0(0.01) ↓0.01| 0.90 (0.92) ↓0.02| 0.04(0.37) ↓0.33| 0.47(0.75) ↓0.28|
> > | Disassemble| 0.78(0.80) ↓0.02| 0.04(0.06) ↓0.02| 0.0(0.03) ↓0.03| 0.02(0.19) ↓0.17| 0.37(0.59) ↓0.22|
> >
> > Cago maintains strong performance even with many failed demonstrations, while the baselines degrade significantly. This robustness to suboptimal or noisy data stems from two key factors:
> >
> > * 1. Cago prioritizes goals that are both challenging and feasible, regardless of whether they originate from successful or failed demonstrations. While failed trajectories may not directly lead to task success, they can expose the agent to diverse regions of the environment, potentially aiding exploration. This allows Cago to extract useful learning signals from a broader range of experiences.
> >
> > * 2. Cago neither imitates demonstrations nor infers reward functions from them. Instead, it uses demonstrations solely to extract sequences of observations that serve as candidate goals. These goals are selected based on the agent’s own learning progress, forming a curriculum that guides policy improvement. This design reduces sensitivity to demonstration quality.
> >
> > We will incorporate these results in the final revision. We thank the reviewer for this suggestion that has improved both the empirical clarity and conceptual scope of Cago.
> >
> > **Addressing Scalability to Abstract or Semantic Goal Spaces**
> >
> > In our implementation of Cago, we use the observation space as the goal space and adopt simple similarity metrics such as MSE or L2 distance to demonstrate the applicability and robustness of Cago's curriculum strategy in both state-based and pixel-based environments. We believe Cago can be naturally extended to support more complex settings. For example, in high-dimensional visual environments, Cago could adopt more expressive similarity metrics such as SSIM (Structural Similarity Index) or FSIM (Feature Similarity Index). In tasks involving semantic or language-conditioned goals, one could compute a scalar score function $\text{Score}(s, g)$ that estimates how well a visual state $s$ satisfies a goal $g$, and then compare states based on the difference in their scores with respect to the same goal. This score function can be instantiated using pretrained vision-language models, allowing goal representations to move beyond raw states to language embeddings or other semantic forms.
> >
> > Our main contribution is a general framework for goal sampling from demonstrations—based on the agent's capability—which can be paired with any goal space and similarity metric appropriate to the domain. We will revise the paper to clarify this design philosophy and emphasize Cago's extensibility. We thank the reviewer for this insightful suggestion!

---

### Official Review · Reviewer_x9ni · 2025-07-02

**Clarity:** 3
**Significance:** 3
**Originality:** 3
**Rating:** 5
**Confidence:** 3

**Summary:**

The paper introduces a novel method to guide learning when demonstrations are available. It suggest to track the progress along the demonstrations by keeping track of visitation counts. Goals are then sampled around the capability limits of the agent to support steady progress. This is implemented as part of a goal-conditioned model-based RL agent and tested across a variety of tasks.

**Questions:**

* Provide a clearer framing, I would suggest committing to the IL narative.
* Extend the related work to better cover imitation learning (e.g. [Reddy2020], [Kostrikov2020] [CSIL2023] [Sikchi2024] from above)
* Provide more IL state-of-the-art baseline.
* Include results on D4RL for which there are results on established baselines
* Provide further discussion on initial state resets, how this impacts real world robot experiments, and what alternatives there are, if any?
* Further details on the implications of theorem 1 would be helpful.

**Ethical Concerns:**

["NO or VERY MINOR ethics concerns only"]

**Final Justification:**

The authors have addressed my concerns. I am increasing my score and I am curious to see the CSIL baseline results.

**Limitations:**

Limitations around environment resets and similarity metric could be further discussed.

**Quality:**

3

**Strengths And Weaknesses:**

The paper has a good structure and reads well.

The core idea of guiding learning by leveraging demonstrations is very interesting and addresses a highly relevant problem in imitation learning. While certain design decision may be debatable (e.g. the choice of a visitation dictionary and the use of a simple similarity metrics), the paper shows a successful application of the idea.

The proposed set of ablations is very interesting and confirms the algorithmic design choices.

Having said that, the framing of the method could be improved. While the method is likely applicable in different setups where demonstrations are available, the motivation and introduction mainly suggest application in imitation learning (using references to GAIL or PWIL). The paper then slightly deviates towards goal-conditioned methods, which still maintain a link to IL and is a sensible choice with suggested goal sampling. However, during evaluation many of the suggested baselines (JSRL, MoDem, CalQL, RLPD) are RL algorithms which leverage demonstrations in conjunction with a reward. As such they cannot directly serve as a imitation learning baseline and I wonder whether the environment reward is used? If there are used as RL backbone in conjunction with a Cago driven imitation reward they would rather represent ablations. On the other hand, there are not many online state-of-the-art imitation learning algorithm present among the baselines except for PWIL. I would suggest sticking with the IL narrative in which case further baselines would be required, e.g. SQIL [Reddy2020], CSIL [Watson2023], ValueDice [Kostrikov2020], or ReCOIL [Sikchi2024].

While there is already a wide set of tasks in the paper I would suggest further evaluating on D4RL [Fu2020] in line with the above baselines.

According to Algorithm 2 $D_{cap}$ is not used to train $\pi^G$. Is this correct? So the distance function and policy are only trained on simulated data? It feels a bit wasteful to use $D_{cap}$ for the world model only. According to the rational behind eq (9) the left term could be attenuated with real world transitions, no? Overall, I would very much like to see the method applied to model-free RL.

The use of the similarity metric to count visitation could be a limitation for more complex tasks, in particular when moving images are involved. But also state spaces with highly varying ranges could probably be challenging. Could you maybe further comment?

While not as bad as arbitrary resets, having to reset to specific initial states remains an extra burden, in particular for real world robots which usually contain non-controllable objects. It would thus be good to better understand the challenge with sampling parts of the initial states away from the demonstrations and whether this could maybe be addressed differently?

I am not sure I fully understand the importance of Theorem 1. The model prediction error in eq (9) is over the full state space, but the theorem bounds it under the visitation of the learned policy and model. Also, it feels like assumption (2) would mean BC is good enough.

[Reddy2020] Reddy et al., SQIL: Imitation Learning via Reinforcement Learning with Sparse Rewards, ICLR 2020.
[Kostrikov2020] Kostrikov et al., Imitation Learning via Off-Policy Distribution Matching, ICLR 2020.
[CSIL2023] Watson et al., Coherent Soft Imitation Learning, NeurIPS 2023.
[Sikchi2024] Sikchi et al., Dual RL: Unification and New Methods for Reinforcement and Imitation Learning, ICLR 2024.
[Fu2020] Fu et al., D4rl: Datasets for deep data-driven reinforcement learning.

---

> ### Author Rebuttal · Authors · 2025-07-31
>
> We appreciate your insightful feedback and constructive comments!
>
> **Q1. Provide a clearer framing, I would suggest committing to the IL narrative.**
>
> We would like to clarify that Cago is fundamentally different from traditional IL methods, both in methodology and motivation. IL typically requires direct learning from demonstrations, either by recovering a reward function from demonstrations, or matching expert and agent state-action distributions (e.g. GAIL), or incorporating demonstrations as anchors to mitigate overestimation and instability caused by out-of-distribution actions (e.g. Cal-QL). In contrast, Cago presents a new paradigm to utilize demonstrations: it dynamically tracks the agent's competence along demonstrated trajectories and uses this signal to select intermediate states in the demonstrations that are just beyond the agent's current reach as goals to guide
> online trajectories collecting. To evaluate this novel perspective, we compared Cago against state-of-the-art baselines that represent diverse strategies for learning from demonstrations, including distribution mathcing (GAIL and PWIL), curriculum learning (JSRL), model-based exploration (MoDem), and offline-to-online fine-tuning (CalQL and RLPD). These comparisons are essential to highlight the effectiveness of Cago as a new paradigm for learning from demonstrations.
>
> ***We clarify that our baselines can access an absolute sparse reward where +1 is rewarded only on success states and 0 is given otherwise. This reward is, however, not accessible to Cago during training.***
>
> **Q2. Extend the related work to better cover imitation learning**
>
> In the revision, we will expand the Related Work section to include a broader and more up-to-date survey of IL literature and clarify more explicitly how Cago differs from standard IL methods.
>
> **Q3. Provide more IL state-of-the-art baseline**
>
> We appreciate the suggestion to include more imitation learning baselines for clarity and completeness. To address this, we conducted additional experiments comparing Cago with two suggested state-of-the-art imitation learning methods—SQIL and ValueDice—in addition to our existing IL baselines, GAIL and PWIL, across three environments: Stick-Push, Disassemble, and Adroit-Pen. The table below reports the final success rates averaged over 5 random seeds (each evaluated over 100 test episodes):
>
> | Task            | Cago (Ours) | SQIL [1] | ValueDice [2] | GAIL | PWIL |
> |-----------------|-------------|----------|----------------|------|------|
> | Stick-Push      |     **0.99**       |    0.78     |      0.48       |  0.02   |  0.94   |
> | Disassemble     |     **0.80**       |    0.34     |       0.18        |  0.05   |  0.02   |
> | Adroit-Pen      |     **0.82**       |   0.44   |       0.24        |  0.15   |  0.21   |
>
> Cago consistently outperforms these baselines. We will update the paper to include the additional baselines suggested by the reviewer across all our benchmark tasks.
>
> **Q4. Include results on D4RL for which there are results on established baselines.**
>
> We would like to clarify that our evaluation already includes tasks from the D4RL benchmark. Specifically, the Adroit environments used in our experiments—Door, Hammer, and Pen—are directly taken from D4RL.
>
> **Q5. Provide further discussion on initial state resets, how this impacts real world robot experiments, and what alternatives there are, if any?**
>
> Indeed, resetting to specific initial states is a limitation of Cago, but as the reviewer notes, it is already a substantial improvement over arbitrary state resets, which are often infeasible in real-world robotics. Importantly, Cago does not require exact resets to the full demonstration initial states. For instance, in the Peg Insertion task, the peg's initial position can differ from the demonstration, and Cago can learn to reach states within the early portion of the demonstration. However, some elements—like the hole and the box—are non-movable and randomly initialized by the environment. Since the agent cannot manipulate these objects, we have to reset them to match the demonstration configuration to ensure goal reachability. The constraint over other controllable components, such as the robot arm and peg, can be relaxed. This partial reset strategy reduces the burden and makes Cago more practical for real-world settings.
>
> **Q6. Further details on the implications of theorem 1 would be helpful. Also, it feels like assumption (2) would mean BC is good enough.**
>
> Theorem 1 is intended to formalize that the imagined rollouts generated by the world model are relevant and close to trajectories that could have been sampled in the real environment. This property is important because it ensures that policies trained over imagined rollouts can generalize to the real world. Assumption (2) states that the world model can be accurately learned via supervised learning from trajectories collected by the BC Explore policy. This is a reasonable assumption, and it does not require the BC Explore policy itself to be of high quality.
>
> **Q7. $D_{cap}$ is not used to train $\pi_{G}$ in Algorithm 2 and it feels a bit wasteful.**
>
> $D_{cap}$ plays a crucial role in training $\pi_{G}$. We learn $\pi_{G}$ based on imagined rollouts from the learned world model $\widehat{\mathcal{M}}$. Each rollout begins at $s_0$, a state randomly sampled from a trajectory $\tau$ in $\mathcal{D}_{cap}$, and is rolled out for $H$ steps using the goal-conditioned policy $\pi^G(a_t | s_t, g)$. The goal state $g$ is selected as a future state $s_H$ from the same trajectory $\tau$. The objective is to train $\pi^G$ to reinforce trajectories that efficiently reach $g$ in the imagined rollouts from $s_0$ under the learned dynamics $\widehat{\mathcal{M}}$.
>
> **Q8. Cago applied to Model-free RL.**
>
> In early experiments, we attempted to train Cago in a model-free manner by learning the goal-conditioned policy directly from real environment rollouts. But we further find that instead of directly using $D_{cap}$ to train $\pi_G$ in Model-free manner, the world model can offer a richer set of imagined rollouts to train the $\pi^G$, which improves the learning efficiency. In our setting, the simulated trajectories have starting states and goal states randomly sampled from the same trajectory in $D_{cap}$. As a result, the simulated trajectories still resemble segments from $D_{cap}$​, while significantly increasing trajectory diversity and data richness. The MBRL—leveraging simulated trajectories allows us to generate more diverse and abundant training data. This promotes generalization across the environment.
>
> **Q9. The use of the similarity metric to count visitation could be a limitation for more complex tasks.**
>
> We have evaluated Cago in visual environments (Appendix F.2) and found it performs comparably to the state-based setting. While that experiment used MSE to compute differences between image-based observations, more advanced metrics—such as SSIM (Structural Similarity Index) or FSIM(Feature Similarity Index)—can be incorporated. Importantly, the similarity metric used in our paper serves as a simple instantiation, rather than a core component of the paradigm. Cago can be configured with alternative metrics, such as similarity between state embeddings in a world model's latent space, to reduce input dimensionality, or use other signals like future state predication uncertainty from an ensemble of world models to assess whether the surrounding space has been sufficiently explored.

---

> > ### Comment · Reviewer_x9ni · 2025-08-03
> >
> > Thank you for the further explanations.
> >
> > Regarding the framing, I understand that goal-based method have some affinity to RL based method. Application wise however, it makes a big difference whether an exogenous reward is required or not, and would thus keep recommending focussing on IL only. Alternatively, Cago could be extended to employ the sparse reward as well if an apples-to-apples comparison was desired.
> >
> > Regarding baselines, looking at the CSIL results, they seem to reach a very high performance as well in the online setup when provided 10 demonstrations for adroit Door and Hammer.
> >
> > Resets and similarity metrics remain prerequisites for the method, but I think this as been sufficiently discussed.

---

> ### Author Response · Authors · 2025-08-06
> **Thank you for your feedback!**
>
> We thank the reviewer for the constructive feedback. Your observation regarding whether exogenous rewards are required is a valuable perspective. In the revision, we will strengthen the discussion of Cago's connection to standard imitation learning methods from this viewpoint in both the introduction and related work. For evaluation, we will move the comparison results with GAIL and PWIL (currently in Fig. 8) into the main paper and additionally include SQIL, ValueDice, and CSIL as relevant baselines across all tasks. We have already evaluated SQIL and ValueDice during the rebuttal and found that Cago outperforms both. Unfortunately, we were unable to run the official implementation of CSIL due to CUDA version constraints (their code requires CUDA 11, while our system only supports CUDA 12), but we expect to resolve this in the final version.
>
> Notably, the CSIL paper reports only normalized scores (relative to a base policy) for Adroit Door and Hammer, rather than absolute task success rates. Their results suggest that CSIL does not significantly outperform BC. In our experiments, BC achieves only 24% and 8% success rates on Door and Hammer, respectively, using 10 demonstrations—even after careful tuning. In contrast, Cago significantly outperforms the BC results, demonstrating its effectiveness in the challenging low-demonstration-data regime.

---

### Official Review · Reviewer_xD15 · 2025-07-02

**Clarity:** 3
**Significance:** 3
**Originality:** 2
**Rating:** 4
**Confidence:** 4

**Summary:**

This paper aims to enable agents to learn from demonstrations using goal-conditioned RL. The proposed method, Cago (Capability-Aware Goal Sampling), maintains the visitation frequencies of observations across demonstrations to identify an agent’s currently reachable observations and generates states around those observations as the goal for policy optimization. Cago also learns an exploration policy and a final goal predictor from demonstration data, as well as a model for transition dynamics from data collected online. All these modules are used in the Go-Explore paradigm to learn the final goal-conditioned policies.

Cago is evaluated on multiple robot environments and compared with model-based RL (Dreamer and MoDem), Jump-Start RL, offline data for online learning (Cal-QL and RLPD), and imitation learning algorithms (GAIL and PWIL). The results demonstrate that Cago outperforms most of the selected baselines. The contributions of each component in Cago are also analyzed.

**Questions:**

1. In Figures 8 and 9, Cago does not consistently or significantly outperform other methods. What are the possible reasons?
2. Cago mainly uses state similarity instead of trajectory similarity to identify an agent’s current capability. Can reaching a state with a trajectory that is very different from the demonstrations indicate that the agent has learned the ideal skills?

**Ethical Concerns:**

["NO or VERY MINOR ethics concerns only"]

**Final Justification:**

The authors' responses have addressed most of my concerns. They provided additional explanations of their method and experiments to improve the clarity and the quality of validation.

**Limitations:**

Yes

**Quality:**

3

**Strengths And Weaknesses:**

# Strengths
1. The paper is written with a clear flow, and one can easily catch the high-level idea of the proposed method.
2. Cago is a comprehensive framework that encompasses several key components, extending beyond the core idea of capability-aware goal sampling.
3. The experimental results demonstrate the effectiveness of Cago.

# Weaknesses
1. The core idea of Cago is to use demonstrations to generate a curriculum and incorporate this curriculum into goal-conditioned RL. Prior work has proposed similar ideas, e.g., [1-4], but the experiments do not compare with them. Instead, the baselines evaluated, although falling into the category of learning from demonstrations, are not the most closely related to the proposed approach.
2. I do not fully get the purpose of Section 2. It includes relevant work, problem formulation, and background knowledge, but all these parts are mixed together. I will recommend restructuring this section.
3. BC Explorer appears to be a crucial component for Cago to work, but its training details are unclear. How is it trained?
4. Cago includes multiple trainable modules, including a world model and a goal predictor. How well does each module need to perform to make Cago work? Or are they all end-to-end trained, so their individual performance does not matter? I would like to see more discussions on this.
5. Following the previous point, although the paper mentions that Cago could generalize beyond the demonstration data, given a small number of demonstrations with complex model structures, it is very likely that the model would overfit to the demonstrations. I would like to see some experimental verification or discussions regarding the generalizability of Cago to unseen tasks.
6. All experiments are run with five random seeds, which may not be sufficient to demonstrate statistical significance.

# References
[1] Yengera, Gaurav, et al. "Curriculum design for teaching via demonstrations: theory and applications." Advances in Neural Information Processing Systems 34 (2021): 10496-10509.

[2] Bajaj, Vaibhav, Guni Sharon, and Peter Stone. "Task phasing: Automated curriculum learning from demonstrations." Proceedings of the International Conference on Automated Planning and Scheduling. Vol. 33. 2023.

[3] Dai, Siyu, Andreas Hofmann, and Brian Williams. "Automatic curricula via expert demonstrations." arXiv preprint arXiv:2106.09159 (2021).

[4] Hermann, Lukas, et al. "Adaptive curriculum generation from demonstrations for sim-to-real visuomotor control." 2020 IEEE International Conference on Robotics and Automation (ICRA). IEEE, 2020.

---

> ### Author Rebuttal · Authors · 2025-07-31
>
> We appreciate your insightful feedback and constructive comments!
>
> **1. Cago should be compared with baselines using demonstrations to generate a curriculum [1-4].**
>
> In our experiments, we included Jump-Start Reinforcement Learning (JSRL, ICML’23), a recent and strong curriculum learning baseline. JSRL pretrains a guide policy on offline data to provide a curriculum of starting states for the RL policy, which significantly simplifies the exploration problem. As the RL policy improves, the effect of the guide-policy is gradually diminished, leading to an RL-only policy that is capable of further autonomous improvement.
>
> Regarding the specific curriculum-based approaches in [1–4]:
>
> [1] constructs a personalized demonstration curriculum by computing task-specific difficulty scores based on the teacher’s and learner’s policies. In contrast, Cago does not require hand-crafted difficulty metrics—instead, it extracts intermediate goals directly from the demonstration, and guides the agent to reach them progressively based on its own learning progress. This avoids the challenge of designing difficulty metrics for complex manipulation tasks, which is non-trivial without expert knowledge. [2] assumes access to a demonstrator policy, either via imitation learning or inverse reinforcement learning. However, our experiments show that standard imitation learning performs poorly on our benchmarks, limiting the effectiveness of such approaches in our domain. [3,4] segment demonstrations and construct a curriculum by resetting episodes to intermediate states along the demonstration trajectory, starting from the end and moving backward as learning progresses. Cago explicitly avoids such state resets, which are often unrealistic in real-world robotic settings without simulator support (Line 83).
>
> To further address this concern, we conducted additional experiments comparing Cago to [2] and [3] on the Fetch-PickPlace and Fetch-Slide environments that they were evaluated on. The tables below report the final success rates after 1M training steps, averaged over 5 random seeds (each evaluated over 100 test episodes):
>
> | Task                     |   Cago(Ours)  |    ACED[3]          | Task phasing[2]   |
> |-------------------------|--------------------|-----------------------|------------------------|
> | Fetch-PickPlace   |        **1.00**         |          0.93           |         0.75          |
> |  Fetch-Slide          |        **0.48**         |             -              |        0.24          |
> (We were unable to run ACED on Fetch-Slide.)
>
> We will incorporate these comparisons in the revised paper.
>
> **2. BC Explorer appears to be a crucial component for Cago to work, but its training details are unclear.**
>
> The BC-Explorer is trained using behavior cloning on a limited set of demonstrations $\mathcal{D}_{\text{demo}}$ (Line 3, Algorithm 2). It is a stochastic policy trained to mimic expert actions by maximizing their log-likelihood. This stochasticity—combined with limited demonstrations and controlled training steps—prevents overfitting and ensures a good balance between exploration and imitation.
>
> **3. Cago includes multiple trainable modules. How well does each module need to perform to make Cago work?**
>
> Cago consists of five main modules: (1) a capability-aware goal sampler, (2) a BC explorer, (3) a goal-conditioned policy, (4) a world model, and (5) a goal predictor. Cago trains the world model and goal-conditioned policy in an end-to-end manner. The world model is updated using real trajectories collected by the goal-conditioned policy, whose goal is set by the capability-aware goal sampler (Algorithm 2, Line 18). Upon goal reaching, a pretrained BC explorer takes over for the remaining time steps in a rollout. Simultaneously, the goal-conditioned policy is trained using imagined rollouts generated by the world model (Line 19). These two components are tightly coupled and co-adapt during training. The goal predictor is trained separately on demonstrations to infer the final goal state from an initial observation. It is used only during evaluation and enables the goal-conditioned policy to generalize to test environments where the goal state is not explicitly provided.
>
> We assess the importance of each component as follows.
>
> (1) To ablate the capability-aware goal sampler, we designed two additional goal sampling strategies:
>
> * Step-based curriculum: This baseline samples goals from demonstrations in proportion to training steps (i.e., current training step / predefined total training steps). However, it does not assess the agent's actual capabilities, and may therefore sample goals that are either too easy or too difficult for the agent at its current learning stage.
> * Final-goal sampling (Line 329): This strategy always selects the final observation from a demonstration in the goal phase of our Go-Explore sampling paradigm, ignoring the agent's current goal-reaching capability. The BC Explorer takes control from the goal-conditioned policy halfway through each rollout. The table below reports the results averaged over 5 random seeds:
>
> | Environment     | Cago (Ours) | Step-based Curriculum | Final-goal Sampling |
> |----------------|-------------|------------------------|---------------------|
> | Adroit-Pen     |    **0.82**         |           0.48             |          0.52           |
> | Stick-Push     |      **0.99**       |            0.92            |          0.68           |
> | Disassemble    |      **0.8**       |           0.04             |           0.02          |
>
> Cago's goal sampling strategy significantly outperforms the alternatives, highlighting its critical role in the overall effectiveness of our approach.
>
> (2) As for BC-Explorer, we provide detailed ablation studies in Section 4 (Line 328). Specifically, Cago-NoExplorer removes the BC Explorer. We also evaluate Cago-RandomExplorer, which replaces the BC Explorer with a uniformly random policy during the exploration phase of our Go-Explore-style rollout strategy. The results are in Fig. 4. The table below reports the final success rates averaged over 5 random seeds (each evaluated over 100 test episodes):
>
> | Environment     | Cago-BC-Explorer (Ours) | Cago-NoExplorer | Cago-RandomExplorer |
> |----------------|-------------|------------------------|---------------------|
> | Adroit-Pen     |   **0.82**         |           0.60             |          0.69           |
> | Stick-Push     |      **0.99**       |            0.72            |          0.79           |
> | Disassemble    |      **0.8**       |           0.61             |           0.72          |
>
> The results demonstrate that both ablations significantly reduce performance, highlighting the importance of BC Explorer for generating relevant exploratory rollouts to accelerate learning.
>
> (3) For the goal-conditioned policy, we observed that its critic loss decreases over time in correlation with the success rates reported in Fig. 3, underscoring its crucial role in overall performance.
>
> (4)  In early development, we attempted a model-free variant of Cago by training a goal-conditioned policy directly from environment rollouts starting at demonstration initial states using actor-critic methods. However, due to the limited number of demonstrations, this approach failed to generalize. We therefore adopted the model-based strategy that leverages a world model to generate diverse imagined rollouts (Line 173), enabling more effective and sample-efficient policy learning.
>
> (5) Empirically, we find that the Goal Predictor reliably infers final goals from held-out demonstrations, enabling effective zero-shot execution in unseen test tasks.
>
> **4. Given a small number of demonstrations with complex model structures, it is very likely that the model would overfit to the demonstrations.**
>
> Cago mitigates overfitting to a small number of demonstrations by training its goal-conditioned policy $\pi^G$ within a world model. Rather than training $\pi^G$ to reach states solely from demonstrations, Cago instructs $\pi^G$ to learn to reach diverse states sampled from real rollouts in $\mathcal{D}_{\text{cap}}$, which records the trajectories generated under our Go-Explore paradigm with capability-aware goal sampling, in its world model. This promotes generalization across the environment. To evaluate the generalization capability, we tested Cago on 500 unseen initial states generated from random seeds, each differing from those in the demonstrations. We report both the average L2 deviation from demonstration initial states and the success rate across these trials.
>
> | Environment     | Average initial observation deviation | Average success rate(500 seeds) |
> |----------------|-------------|------------------------|
> | Adroit-Pen     |      1.2749     |          0.824              |
> | Stick-Push     |     0.0547        |           0.982             |
> | Disassemble    |      0.1405       |            0.806           |
>
> The results are consistent with Fig 3.
>
> **5. Can reaching a state with a trajectory that is very different from the demonstrations indicate that the agent has learned the ideal skills?**
>
> Cago adopts a curriculum-style sampling strategy, where goals are selected progressively from easier to harder states along the demonstration trajectories. This naturally encourages the agent to incrementally reproduce and internalize the demonstrated behaviors, as illustrated in Fig.1 (Right) and Fig.4(d).
>
> **6. Cago does not consistently or significantly outperform other methods.**
>
> ShelfPlace is the only environment where Cago is outperformed by GAIL. In all other environments, Cago either outperforms the baselines or achieves near 100% success. On ShelfPlace, we observe that GAIL learns a shortcut to achieve the goal, whereas Cago follows the demonstration more faithfully, which results in slower task completion.

---

> > ### Comment · Reviewer_xD15 · 2025-08-04
> >
> > Thank you for the detailed responses. The additional explanations and experiments have addressed some of my concerns.
> >
> > However, my biggest concern is that there are only five random seeds run in each experiment, and from all the learning curves that include variance of the results, I would not consider that as absolutely outperforming other baselines.
> >
> > I have read through others’ reviews and authors’ responses. After careful consideration, I decided to raise my rating due to the completion of the framework and the additional analyses. But still, I believe the method should be more rigorously validated.

---

> ### Comment · Area_Chair_4hTj · 2025-08-04
> **Sufficient response?**
>
> Dear reviewer,
>
> Have the authors assuaged your concerns? Would you be able to move to a positive score?

---

> ### Author Response · Authors · 2025-08-06
> **Thank you for your feedback!**
>
> We thank the reviewer for reinforcing this concern. To further strengthen the reliability of our results, we conducted additional experiments using 3 more random seeds on three representative environments: Adroit-Pen, Stick-Push, and Disassemble. This brings the total to 8 seeds per environment. We compared the performance of Cago against the best-performing baseline. The final success rates and standard deviations at 1M steps are summarized below:
>
> | Environment| Ave success rate (Cago 5 seeds) | std (Cago 5 seeds) | Ave success rate (Cago 8 seeds) |std (Cago 8 seeds)| Best baseline success rate| Best baseline std |
> |----------------|-------------|------------------------|---------------------|---------------------|---------------------|---------------------|
> | Adroit-Pen     |    **0.82**         |          0.177             |         0.813           |       0.145      |     0.66 (RLPD)    |    0.218   |
> | Stick-Push     |      0.99       |           0.01            |         **0.994**          |       0.009       |      0.98 (RLPD)     |  0.021     |
> | Disassemble    |      0.8       |           0.148            |          **0.815**          |      0.125    |    0.59 (Modem)    |      0.097   |
>
>
> These results show that the additional runs yield means similar to those reported with 5 seeds, while further reducing the standard deviation, indicating improved statistical confidence. We hope this extended evaluation more clearly demonstrates the robustness of our method.
>
> In the final version, we will evaluate Cago and all baselines using 10 random seeds to further strengthen the empirical findings. Thank you for your thoughtful comment to help improve our paper!

---

> > ### Comment · Reviewer_xD15 · 2025-08-07
> >
> > Thank you for conducting additional experiments! It would help demonstrate the statistical significance, given that RL results can be largely affected by the random seeds used. I will maintain my current positive rating.

---

### Note · Authors · 2025-08-12

We thank the reviewers and the AC for their engagement and insightful feedback, which have greatly improved the clarity and scope of our work. The main concerns raised and addressed in our responses include:

(1) Importance of the individual modules [Reviewer xD15, NtvZ]: We conducted thorough ablation studies assessing each component's contribution, demonstrating that the capability-aware goal sampler, BC-Explorer, and world models are all critical for Cago's success.

(2) Demonstration quantity [Reviewer xD15,NtvZ]: Designed for the low-demonstration regime, Cago was evaluated with only 10–20 demonstrations to emphasize sample efficiency and generalization. Ablations in the supplement show performance scales with 5, 10, and 20 demonstrations. We also tested with 50 demonstrations during the rebuttal period, where Cago achieves higher success rates and better generalization but slower warm-up. Cago mitigates overfitting to a small number of demonstrations by training goal-conditioned policies inside a learned world model. Evaluations on unseen initial states confirm strong generalization beyond the demonstration distribution.

(3) Robustness to suboptimal demonstrations [Reviewer 1nz5]: While our theoretical guarantees assume high-quality demonstrations, experiments on Adroit environments include suboptimal trajectories, where Cago remains effective. We conducted additional experiments with noisy, unsuccessful demonstration trajectories. Cago maintains strong performance, outperforming baselines that degrade significantly. This robustness arises because Cago extracts sequences of observations from demonstrations as curriculum goals without directly imitating demonstrations or inferring reward functions, reducing reliance on demonstration quality.

(4) Imitation learning (IL) framing [Reviewer x9ni]: We will strengthen Cago's connection to standard IL in the revision. Comparisons with GAIL and PWIL will be moved from the appendix to the main paper, and SQIL, ValueDice, and CSIL will be added as baselines. Our rebuttal results show Cago outperforms these baselines.

(5) Statistical significance [Reviewer xD15]: Additional runs during the rebuttal phase confirm results consistent with our original 5 seeds and reduced variance. We will evaluate all methods using 10 seeds in the revision to strengthen empirical confidence.

We will incorporate these ablations and analyses, along with clarifications addressing other reviewer concerns, in the final version.

---

### Decision · Program_Chairs · 2025-09-17

**Decision:**

Accept (poster)

**Comment:**

This paper presents an empirical method for inverse reinforcement learning in goal-condition RL, whch combines behavioral cloning with a generative moel, and evaluates it in sparse reward environments. The setting is not pure IRL, as the agent uses the generative model to obtain more data using a policy of its choice.

+ The performance is generally strong
- The method is somewhat adhoc, but it makes intuitive sense to some extent.
- Since this is not pure IRL, it would make sense to compare with other methods combining IRL with RL

All reviewers are for acceptance.